# The pan-genome and local adaptation of *Arabidopsis thaliana*

Minghui Kang [1,2,4], Haolin Wu[2,4], Huanhuan Liu [2,4], Wenyu Liu[1], Mingjia Zhu[1], Yu Han[2], Wei Liu[2], Chunlin Chen[2], Yan Song[2], Luna Tan[2], Kangqun Yin[2], Yusen Zhao[2], Zhen Yan[2], Shangling Lou [1,2] ✉, Yanjun Zan [3] ✉ & Jianquan Liu [1,2] ✉

*Arabidopsis thaliana* serves as a model species for investigating various aspects of plant biology. However, the contribution of genomic structural variations (SVs) and their associate genes to the local adaptation of this widely distribute species remains unclear. Here, we de novo assemble chromosome-level genomes of 32 *A. thaliana* ecotypes and determine that variable genes expand the gene pool in different ecotypes and thus assist local adaptation. We develop a graph-based pan-genome and identify 61,332 SVs that overlap with 18,883 genes, some of which are highly involved in ecological adaptation of this species. For instance, we observe a specific 332 bp insertion in the promoter region of the *HPCA1* gene in the Tibet-0 ecotype that enhances gene expression, thereby promotes adaptation to alpine environments. These findings augment our understanding of the molecular mechanisms underlying the local adaptation of *A. thaliana* across diverse habitats.

*Arabidopsis thaliana* (2n = 10) (Brassicaceae) has been used as a model plant across many studies because of its small genome size, short generation time, and the large number of seeds produced from each mother plant. In addition, due to its worldwide distribution covering habitats with extensive ecological diversity throughout Eurasia, Africa, and North America, *A. thaliana* is also ideal species for revealing molecular mechanisms of ecological adaptation in plants[1]. In 2000, the *A. thaliana* genome, based on the Col-0 ecotype, was the first completely sequenced and assembled plant genome; this work has greatly advanced molecular studies[2]. With the continued advancement of sequencing technology, four versions of an *A. thaliana* Col-0 ecotype telomere-to-telomere (T2T) reference genomes have been published and updated[3-6] Furthermore, the genomes of several other *A. thaliana* ecotypes have also been published[7].

Population genomic analyzes based on the reference genome and whole-genome resequencing data of other ecotypes have revealed a widespread global postglacial expansion of *A. thaliana* from sparsely distributed relict ecotypes[8]. In particular, a large number of genetic variations were associated with multiple phenotypic changes underlying *A. thaliana* ecological adaptation to varied habitats[8]. These genetic variations across ecotypes are mainly comprised of single nucleotide polymorphisms (SNPs) and short insertions and deletions (INDELs, often <50 bp)[9,10]. Allelic variations in major genes associated with ecological phenotypes have also been uncovered through genome-wide association studies (GWAS)[11,12]. Beyond SNPs and INDELs, there are only a few studies on whether variable genes and large structural variations (SVs, often > 50 bp) contribute to ecological adaptation[13]. The SVs are mainly comprised of presence/absence variants, inversions, translocations, and copy number variations. These SVs may affect gene expression and sometimes can remove existing genes and produce new genes. Some evidence suggests SVs are important contributors to phenotypic variation[13,14]. However, incomplete detection of genomic variants may lead to weak linkage disequilibrium (LD), which may have decreased the statistical power of previous GWAS analyzes that have ultimately failed to identify the major genetic loci underlying ecological phenotypes[15,16]. Assembling

[1]State Key Laboratory of Grassland Agro-ecosystem, College of Ecology, Lanzhou University, Lanzhou 730000, China. [2]Key Laboratory of Bio-resource and Eco-environment of Ministry of Education, College of Life Sciences, Sichuan University, Chengdu 610065, China. [3]Key Laboratory of Tobacco Improvement and Biotechnology, Tobacco Research Institute, Chinese Academy of Agricultural Sciences, Qingdao 266000, China. [4]These authors contributed equally: Minghui Kang, Haolin Wu, Huanhuan Liu. ✉e-mail: shanglinglou@126.com; zanyanjun@caas.cn; liujq@nwipb.ac.cn

high-quality de novo genomes of multiple ecotypes[7,17] and conducting pan-genome analyzes[18,19] of these genomes could reveal SVs and capture previously missing heritability[16]. In addition, a graph-based pan-genome assembly can efficiently integrate genetic variants of all de novo genomes and identify the major SVs underlying diverse phenotypes[20–22].

In this study, we assemble 32 high-quality genomes of representative *A. thaliana* ecotypes from across their respective distributions using PacBio-HiFi long-read sequencing. While some *A. thaliana* ecotypes may be paraphyletic in origin, most of the Eurasian ecotypes likely originate from one recent monophyletic expansion[23]. The 32 select ecotypes include six distinctly districted relict ecotypes, one from the Qinghai-Tibet Plateau of western China, one from Italy, and four from Morocco[23]. The other 26 ecotypes are selected from the monophyletic Eurasian postglacial expansion lines[8] (Supplementary Table 1). These ecotypes cover most major clades and subclades of the 1135 global *A. thaliana* accessions and are representative of the diverse habitats occupied by *A. thaliana*[8]. We recover highly variable genes between ecotypes and also many SVs that are involved in the local adaptation of each ecotype. Our study provides a set of high-quality genetic resources that improve our understanding of the genomic diversity and evolution underlying the ecological adaptation of *A. thaliana*. Additionally, we provide functional tests to confirm the role of SVs and variable genes in the formation of special ecological phenotypes.

## Results

### Chromosome-level genome assemblies and annotation of 32 ecotypes

In order to obtain the genome diversity across different *A. thaliana* ecotypes, we selected 32 representative ecotypes from Europe, Asia, Africa, and North America (including 6 relict ecotypes) for de novo genome assemblies (Fig. 1a and Supplementary Table 1). We generated 2.18 – 8.28 Gb (approximately 15-60 X) high-fidelity (HiFi) reads for the 32 ecotypes (Supplementary Table 2) which we then assembled into contigs using hifiasm and anchored onto the five chromosomes using RagTag with the recently published Col-PEK T2T genome as a reference[6]. We produced and downloaded RNA data for these ecotypes (Supplementary Tables 3 and 4) and estimated genome size for the Col-0 ecotype (Supplementary Fig. 1 and Supplementary Table 5). The final assembly sizes ranged from 129.4 to 144.9 Mb with contig N50 sizes of 5.91 – 20.3 Mb (Supplementary Table 6). The completeness of assemblies was evaluated by Benchmarking Universal Single-Copy Orthologs (BUSCO)[24], with completeness scores of 99.0 to 99.3% (single-copy and duplicated) in the chromosome-scale assemblies (Supplementary Fig. 2 and Supplementary Table 6). The evaluations indicated high contiguity and high completeness of the 32 *A. thaliana* genome assemblies.

Combined with transcriptome-based, ab initio, homologous-protein-based prediction, and gene lift-over using the Araport11 gene annotation file[25], we predicted 27,239 to 28,735 protein-coding genes in the 32 assembled genomes (Supplementary Table 7). Between 481 and 5189 genes were found to have structural differences between ecotype genomes relative to the Araport11 reference, with the differences between relict ecotypes and the Araport11 reference being significantly greater than those between non-relict ecotypes and the reference (Supplementary Fig. 3, 4 and Supplementary Table 8). The completeness of the gene annotations was also evaluated using BUSCO, resulting in completeness scores ranging from 98.9% to 99.7%, suggesting high gene annotation quality (Supplementary Fig. 5 and Supplementary Table 7). Throughout the ecotype genomes, approximately 92.6% to 94.2% of the genes were functionally annotated through at least one database in eggnog[26] (Supplementary Table 7).

To infer the evolutionary relationships of the 32 genomes, we clustered the annotated genes into gene families with the sister species

*A. lyrata* as an outgroup. We selected 17,183 single-copy gene families among these 33 genomes to construct a maximum likelihood phylogeny. The non-relict ecotypes clustered into one monophyletic clade. However, the relict ecotypes were paraphyletic with the Tibet-0 ecotype, which was basal to all other ecotypes (Fig. 1b).

### Pan-genome analyzes

We constructed a gene-family-based pan-genome of the 32 ecotypes by clustering 887,723 genes into 31,318 pan-gene clusters (including 2072 clusters with only one gene) using OrthoFinder with the Markov clustering algorithm. Pan-genome size increased with the number of genomes and approached a plateau (newly added gene clusters number increased by less than 1% with additional added genomes) as n approached 26 (Fig. 1c). Based on the frequency of occurrence of gene clusters in each genome, we classified gene clusters into four categories: 21,545 (68.8%) gene clusters were present in all 32 ecotype genomes and were defined as core gene clusters; 3743 (12.0%) gene clusters appeared in 26 to 31 ecotype genomes and were defined as softcore gene clusters; 3929 (12.6%) gene clusters were found only in 2 to 25 genomes were defined as dispensable gene clusters; and 2101 (6.7%) gene clusters were found only in a single ecotype and were defined as private gene clusters (Fig. 1d).

Gene ontology (GO) term enrichment analysis revealed that the core genes were enriched in basic, critical functions such as flower development, RNA binding, transcription regulation, transport, and cellular homeostasis, which suggests that the core genes are mainly involved in maintaining the basic activities of *A. thaliana* (Fig. 1e). Variable genes (including softcore, dispensable and private genes) were enriched in secondary metabolic processes, cell differentiation, and responses to stresses (Fig. 1f). Private genes were significantly enriched in response to multiple types of stressors such as endogenous stimuli and light stimuli (Supplementary Fig. 6). Further investigation of the associations between the variable genes in the 32 ecotype genomes and the 19 BIOCLIM environmental variables[27] revealed that mean diurnal range (BIO2) and temperature annual range (BIO7) were significantly associated with the presence/absence of variable genes (Supplementary Fig. 7 and Supplementary Table 9). Functional enrichment analysis of 215 variable gene families significantly associated with BIO2 and BIO7 showed that these genes were also enriched in response to different types of stress (Supplementary Fig. 8). These results suggest that the variable genes are likely associated with adaptation to ecotype-specific local environments.

Gene expression analysis showed that the variable genes had lower expression levels than the core genes. In addition, the non-synonymous/synonymous substitution ratio (Ka/Ks) analysis indicated that variable genes had higher pairwise Ka/Ks values than the core genes (Fig. 1g, h). These results suggest that the function of core genes is relatively conserved across ecotypes, while variable genes evolve more rapidly to obtain new functions or adapt to the new environment, or the difference in Ka/Ks could simply be relaxed selection on non-core genes.

### The transposable elements (TEs) landscape of 32 *A. thaliana* genomes

We constructed a pan-TE library for the 32 *A. thaliana* genomes using Repbase and EDTA de novo TE annotation and obtained 780 non-redundant TE families (Supplementary Table 10). Then, we annotated TEs in each genome using RepeatMasker and the constructed pan-TE library. The annotation classified the 780 TE families into three categories based on their frequency of occurrence in each genome: core TEs (present in all 32 genomes), variable TEs (present in 6-31 genomes), and rare TEs (present in 1-5 genomes) (Fig. 2a). In all TE families, DNA transposons (26% of TEs) and long terminal repeat-retrotransposons (LTRs; 62% of TEs) accounted for the majority of TEs. In addition, variable TEs were mainly of the LTR type (Fig. 2b, c).

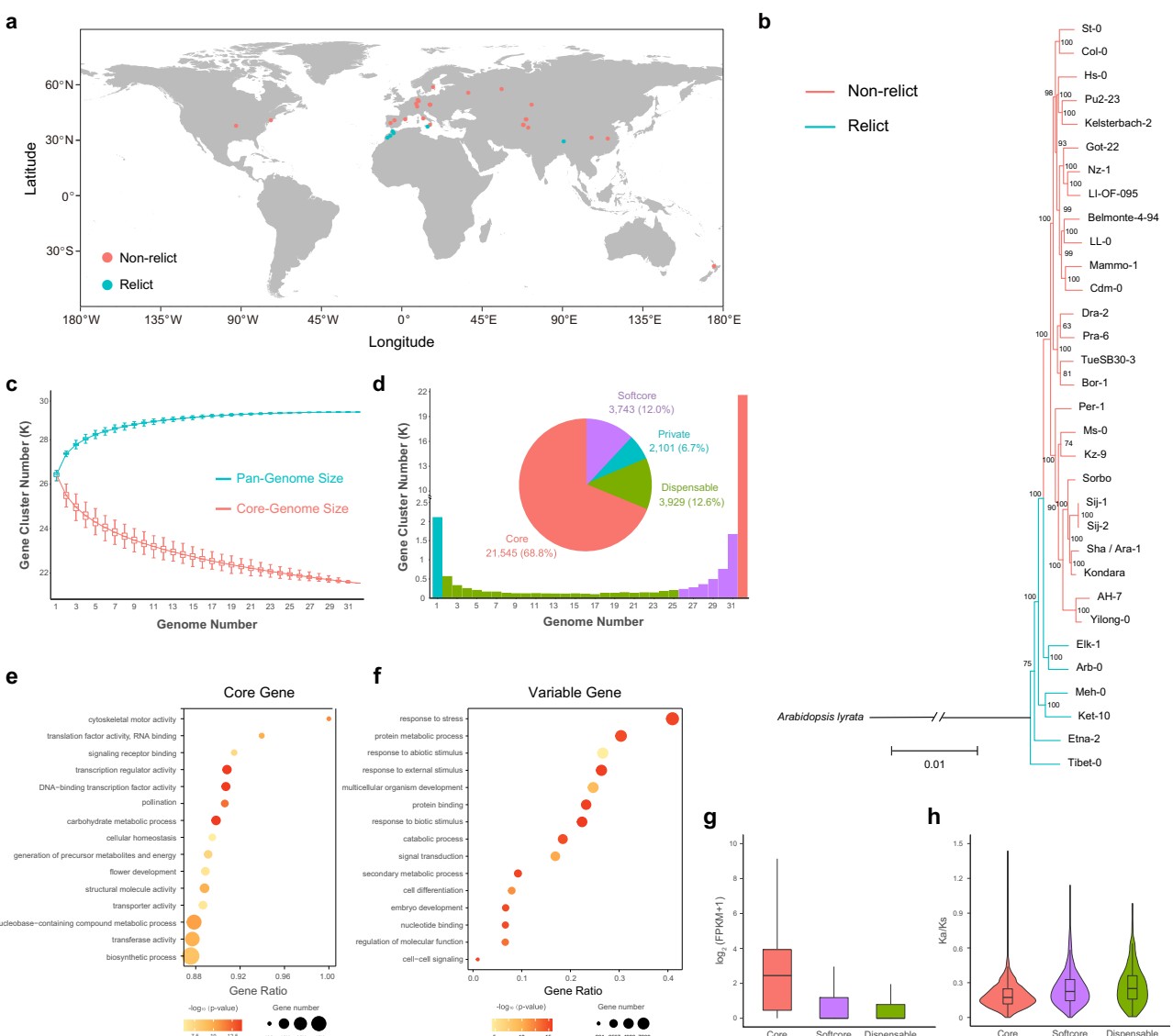

**Fig. 1 | Pan-genome of 32 A. thaliana ecotypes. a** Geographic distribution of 32 selected ecotypes of *A. thaliana*. The red circles represent non-relict ecotypes while the blue circles represent relict ecotypes. **b** Phylogenetic tree of 32 *A. thaliana* ecotypes with *A. lyrata* as the outgroup. Bootstrap values (%) are displayed on each branch. The red branches represent non-relict ecotypes while the blue branches represent relict ecotypes. **c** Pan-genome and core genome size simulated by gene cluster number and pan-genome composition. The upper and lower edges of the boxes represent the 75% and 25% quartiles, respectively, while the whiskers extend to 1.5× the inter-quartile range (IQR). The sample size was set to 1000 and the sample repeat was set to 30. **d** Number and percentage of core, softcore, dispensable, and private gene clusters. **e** Bubble chart

of gene ontology (GO) enrichment analysis for core genes. Significance was tested by two tailed Fisher's exact test method. **f** Bubble chart for the GO enrichment analysis of variable genes. Significance was tested by two tailed Fisher's exact test method. **g** Expression levels of genes belonging to core ($n = 709,766$), softcore ($n = 125,555$), and dispensable ($n = 50,237$) gene families. **h** Pairwise nonsynonymous/synonymous substitution ratios (Ka/Ks) within core ($n = 709,766$), softcore ($n = 125,555$), and dispensable ($n = 50,237$) genes. The upper and lower edges of the boxes represent the 75% and 25% quartiles, the central line denotes the median, and the whiskers extend to 1.5× IQR in **g** and **h**. Source data are provided as a Source Data file.

The TE content also varied between ecotypes, which ranged from 20.34% to 26.44% of each genome (Supplementary Table 6). This variable content among genomes led to differences in genome size among ecotypes (Fig. 2d and Supplementary Fig. 9). Among all TE categories, LTRs and DNA transposons (such as terminal inverted repeats; TIR) were the two most abundant categories across genomes (Fig. 2e). Furthermore, we identified the intact LTRs across ecotypes (Supplementary Table 11) and estimated their insertion times. We found that most of the intact LTRs across genomes expanded within the last one million years, though numerous LTRs in non-relict ecotypes originated more recently (Fig. 2f and Supplementary Fig. 10). This may have led to the emergence of new LTR families and the variance of LTR families between relict and non-relict ecotypes.

To evaluate the effect of TE insertion on gene expression, we compared the gene expression levels of genes with and without TE insertion (TE overlapping with gene region). Genes with inserted TEs displayed lower expression levels (Fig. 2g). GO enrichment analysis showed that the TE-inserted genes were mainly enriched in cell–cell signaling, lipid metabolic processes, and response to stressors, including biotic and external stimuli (Supplementary Fig. 11). For example, *CCR1* (AT1G15950) encodes a cinnamoyl CoA reductase involved in lignin biosynthesis and cell proliferation in leaves. The *ccr1* mutants exhibit increased ferulic acid (FeA) content, which has antioxidant activity and reduces the levels of reactive oxygen species (ROS) in plants[28]. Across 32 ecotypes, we found a specific DNA/MULE-MuDR insertion that occurred in the intron region of *CCR1* in only two

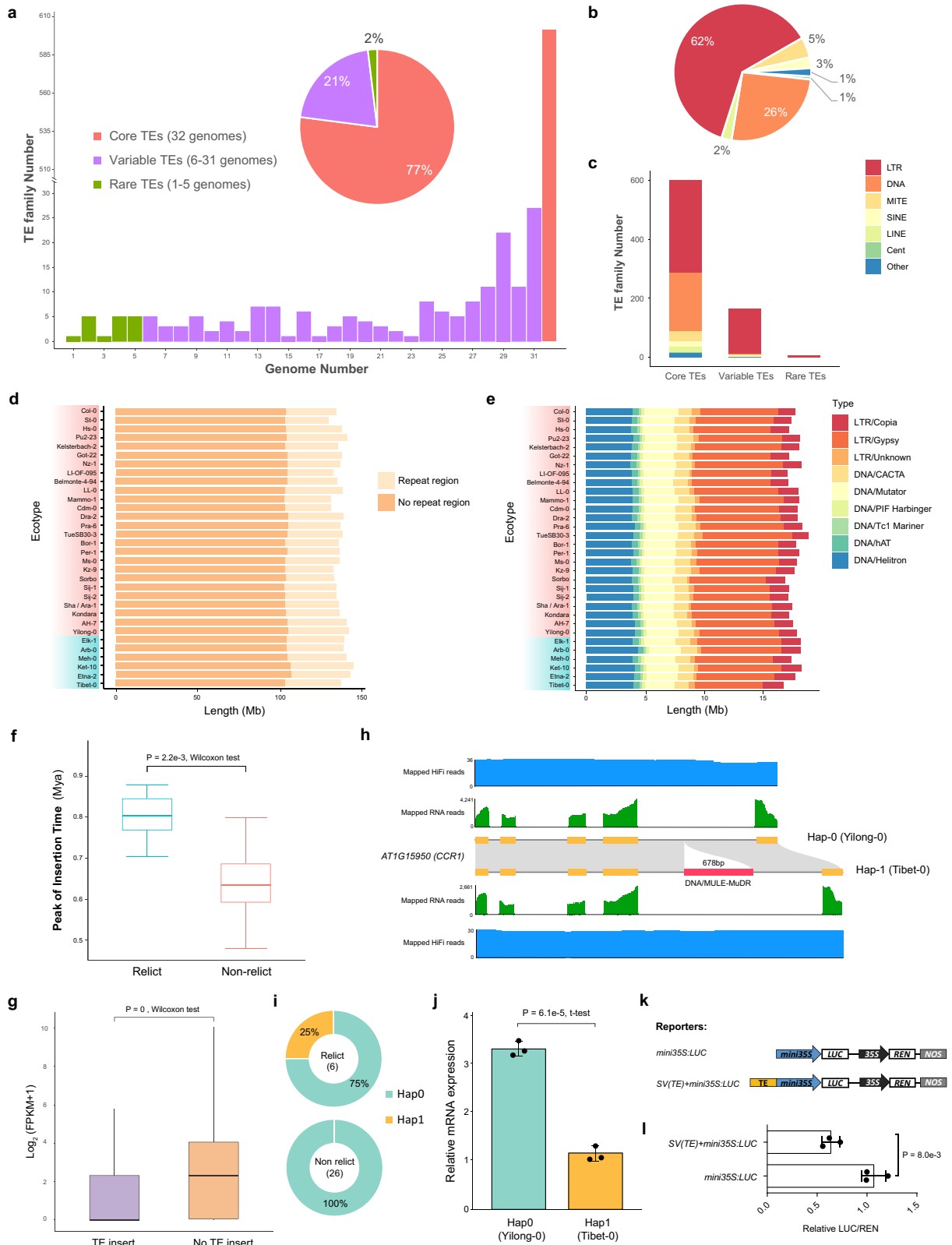

relict ecotypes (Tibet-0 and Meh-0). This insertion reduced the expression of *CCR1*, which was confirmed using in vivo dual-luciferase (Dual-LUC) activity assays (Fig. 2h–j and Supplementary Fig. 12). Both ecotypes with this insertion mutation occur in arid habitats[23,29], and we speculate that reduced CCR1 expression may have promoted the adaptation of both ecotypes to arid habitats through increasing anti-oxidant activity while reducing ROS.

We also studied the types and locations of TEs inserted around genes and their influence on gene expression. TEs tended to be inserted into variable genes (33.26%, 2561/7701), while the proportion of TE insertions in core genes were comparatively smaller (15.28%, 3292/21545). DNA transposons were the most frequent type of TE insertion, followed by the LTR type. Among genes with putative functional enrichments for habitat adaptation, TEs were more likely to

**Fig. 2 | Repetitive sequences of 32 de novo genomes. a** Number and percentage of core, variable, and rare transposable element (TE) families. **b** Classification of 780 pan-TE families. **c** Distribution of TE types in core, variable, and rare TE families. **d** TE length identified in different *A. thaliana* genomes. **e** Composition of different TE types in *A. thaliana* genomes. Blue rectangles display relict ecotypes while red rectangles display non-relict ecotypes. **f** Comparison of peak intact long terminal repeat-retrotransposons (LTR) insertion times of relict ecotypes (*n* = 6) and non-relict (*n* = 26) ecotypes. Significance was determined using a two tailed Wilcoxon test with *p* = 2.2e-3 < 0.05. **g** Comparison of the expression levels between genes with (*n* = 97,922) and without (*n* = 789,801) TE insertion. Significance was determined using a two tailed Wilcoxon test with *p* = 0 < 0.05. The upper and lower edges of the boxes represent the 75% and 25% quartiles, the central line denotes the median, and the whiskers extend to 1.5× inter-quartile range (IQR), and the outliers

are removed in (**f**) and (**g**). **h** The two haplotypes of *CCR1* are determined by the presence or absence of DNA/MULE-MuDR insertion (red bar) in the fourth intron. HiFi and RNA-seq read mapping supports the gene structure annotation. **i** The distributions of the two *CCR1* haplotypes in relict and non-relict ecotypes. **j** Relative *CCR1* mRNA levels assessed by quantitative RT-PCR. Data are mean ± SD from independent biological replicates (*n* = 3). Significance was determined using a two tailed t-test with *p* = 6.1e-5 < 0.05. **k** Schematic diagram of reporters of transient dual-luciferase assay. **l** Transient dual-luciferase assay in *N. benthamiana*. LUC: Firefly Luciferase; REN: Renilla Luciferase; NOS: NOS terminator. Data are mean ± SD from independent biological replicates (*n* = 3). Significance was determined using a two tailed t-test with *p* = 8.0e-3 < 0.05. Source data are provided as a Source Data file.

insert into the upstream regions of the genes (Supplementary Figs. 11 and 13). The expression level of genes with TE insertion decreased the most when the insertion was in the coding sequence (CDS) region, and among TE types, the LTR type had the greatest impact on gene expression (Supplementary Figs. 14 and 15).

The observed bias in the distribution of TE insertions may be attributed to two possible reasons: 1) The initial TE insertions may be random, and their retentions are selected due to the regulation of gene expression with positive adaptive roles. 2) The targeting of TEs could be influenced by specific chromatin signatures. For example, a previously published study demonstrated that the histone variant H2A.Z has a crucial role in the preferential integration of Ty1/Copia retrotransposons into environmentally responsive genes, while avoiding essential genes[30]. These two hypotheses may jointly and non-exclusively affect the distribution of TE insertions in *A. thaliana*. In addition, the type and location preference of TE insertions may be related to the differential expression of genes in different ecotypes, which further promotes adaptation to different environments.

## Graph-based pan-genome and structural variations (SVs) identification

To identify structural variations across 32 ecotypes, we constructed a graph pan-genome by integrating variants from the Minimap2 alignment with Col-0 as the reference (Supplementary Fig. 16). The graph pan-genome comprised a total of 243.27 Mb with 468,168 nodes (the number of fragments of sequences) and 649,692 edges (the connections between nodes). Among them, 203,747 non-reference nodes were identified, accounting for 108.90 Mb of the map genome. The new sequences in each ecotype compared with the Col-0 reference genome varied from 56.58 Kb to 8.45 Mb and had 174 to 49,675 specific edges to connect them to the reference nodes (Supplementary Table 12). On average, each node spanned 0.52 Kb and was connected by 1.39 edges. Based on the sequence of the graph genome, we calculated the pan-genome size and core-genome size (Fig. 3a). The pan-genome size increased with the number of genomes added.

We detected SVs in the graph-based genome using gfatools using the bubble-popping algorithm. After filtering out all SVs less than 50 bp in length, a total of 61,322 SVs were detected in at least one genome as compared with the reference genome (Fig. 3b and Supplementary Table 13). The majority (72.96%; 44,741/61,322) of called SVs were smaller than 500 bp (Supplementary Fig. 17). SVs were further classified into two types: biallelic (with only one non-reference path) and multiallelic (with more than one non-reference path). The biallelic SVs were further divided into insertion, deletion, and divergent types according to the reference paths (including traditional types of SVs: inversion (divergent), translocation (one insertion and one deletion) and duplication (one insertion)) (Fig. 3c). Among the biallelic SVs, the divergent type was the most abundant, with a combined length of 7.51 Mb, while the insertion and deletion types had total lengths of 5.54 Mb and 1.09 Mb, respectively. In addition, the multiallelic type was the largest type of the SVs (29.65 Mb), which

suggests complex SVs exist between different ecotypes (Fig. 3b and Supplementary Table 13).

Among detected SVs, more than 13,913 (22.69%) were correlated with inserted TEs, with the biallelic-insertion type accounting for the largest proportion of inserted TEs (57.4%) (Supplementary Table 13). Most SVs with inserted TEs were larger than 500 bp (60.2%, 8376/ 13,913), accounting for 50.52% of all SVs above 500 bp. In contrast, only 12.38% of SVs below 500 bp had TE insertions. This result suggests that large SVs likely resulted from TE transposition. In addition, the number of SVs was larger in relict ecotypes, and the relict ecotypes had a larger number of specific SVs than the non-relict ecotypes derived from postglacial expansion, suggesting distinct differentiations (Supplementary Fig. 18 and Supplementary Table 14). The Tibet-0 ecotype had the largest number of specific SVs across all analyzed ecotypes. However, SVs in non-relict ecotypes had larger TE insertion proportions, which may be related to the more recent TE expansion mentioned above (Supplementary Table 14).

We next found the intersection of genes annotated in the Col-0 reference genome with the SV regions. We found 7% to 48% of genes and their promoter region (2 Kb upstream from the transcription start site (TSS)) are affected by four types of SVs (including 3415 out of 7701 (44.34 %) of variable gene families mentioned above) and the expression levels of these SV-overlapped genes were significantly decreased compared to those without SVs (Fig. 3d, e and Supplementary Table 13). SVs were more likely to occur in the gene flanking region, and biallelic-divergent SVs affected the largest number of genes (Supplementary Table 15). In addition, the expression level of genes with SVs overlapping the CDS region were significantly lower, but the overlapping SVs type had little effect on gene expression (Supplementary Figs. 19, 20). Functional enrichment analysis showed that SV-overlapped genes were mainly enriched in secondary metabolic processes, enzyme regulator activity, and responses to diverse stressors (Supplementary Fig. 21). In addition, GO enrichment results for genes in SV hotspot regions (SV density in the top 5%) showed an enrichment of genes related to catalytic activity and response to light stimuli (Fig. 3f and Supplementary Fig. 22). Therefore, the widely distributed variable SVs (Fig. 3f) and their overlapped genes may partly account for the ecological adaptation of different ecotypes across diverse habitats.

As an example of SV-overlap influencing adaptation, *KNAT3* (*AT5G25220*) is a class II knotted1-like gene that uses BLH1 to directly regulates *ABI3* expression to modulate seed germination and early seedling development. The *knat3* mutants are less sensitive to ABA or salinity exposure during seed germination with early seedling development[31]. In addition, *KNAT3* was identified to promote secondary cell wall biosynthesis in xylem vessels together with *KNAT7*. The *knat3 knat7* double mutants had reduced stem tensile and flexural strength compared with wild-type and single mutants[32]. Across 32 ecotypes, we revealed an SV in the promoter region of *KNAT3* specific to the relict Tibet-0 ecotype sampled in the high-altitude Qinghai-Tibet Plateau (Fig. 3g, h and Supplementary Fig. 23). The *KNAT3* gene expression level in Tibet-0 was significantly increased compared with

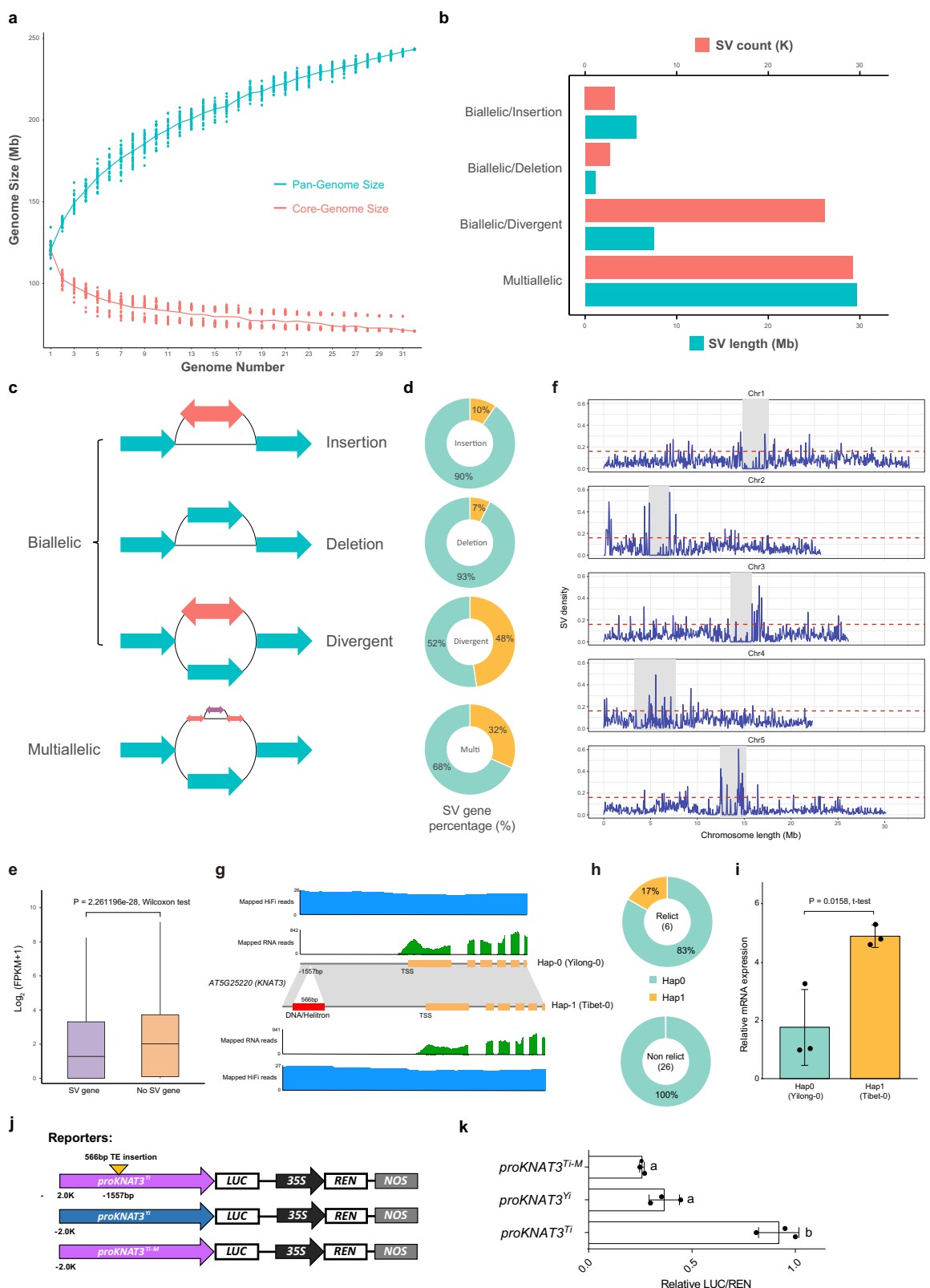

other ecotypes without this insertion, and in vivo Dual-LUC activity assays also confirmed this expression effects because of the 566 bp TE insertion (Fig. 3i–k). The expression level of *KNAT3* was regulated by light as its promoter responded differently to red and far-red light[33]. Therefore, the inserted SV in the *KNAT3* promoter with increased expression level in Tibet-0 may play an important role in its adaptation to the strong light radiation of the high-altitude region.

This expression difference because of the biallelic SVs was also confirmed for two other genes, *WH1 (AT1G54260)* and *HPCA1 (AT5G49760)*. A 180 bp insertion was identified in the promoter region of the *WH1* gene, which was predominantly present in the relict ecotypes compared with the others (Fig. 4a, b and Supplementary Fig. 24). This insertion was found to be associated with the reduced transcriptional expression of *WH1* and increased resistance to UVB stress in one

**Fig. 3 | Characterization of the graph genome across 32 de novo genomes of A. thaliana. a** The graph pan-genome size changes with the increase in number of genome assemblies. **b** The bar chart shows the number (red) and length (blue) of each type of structural variation (SV) separately. **c** Schematic illustration of diverse SV types from the graph pan-genome based on the reference genome Col-0. **d** The pie chart shows the number of genes affected by SV as a proportion of the overall number of genes. **e** Expression levels of SV-overlapped genes (n = 18,883) and non-SV-overlapped (n = 9852) genes. The upper and lower edges of the boxes represent the 75% and 25% quartiles, the central line denotes the median, and the whiskers extend to 1.5× the inter-quartile range (IQR). Significance was determined using a two tailed Wilcoxon test with p = 2.261196e-28 < 0.05. **f** SV density along each chromosome based on Col-0 genome assembly: (50 Kb sliding windows with a step-size of 20 Kb in blue). Gray rectangles: centromeres. The dashed red lines indicate

thresholds for SV density values of in the top 5%. **g** Two haplotypes of *KNAT3* are determined by the presence or absence of DNA/Helitron insertion (red bar) in the promoter region. HiFi and RNA-seq read mapping supports the gene structure annotation. TSS: transcription start site. **h** The distributions of the two haplotypes of *KNAT3* in relict and non-relict ecotypes. **i** Relative *KNAT3* mRNA levels as assessed by quantitative RT-PCR. Data are mean ± SD from independent biological replicates (n = 3). Significance was determined using a two tailed t-test with p = 0.0158 < 0.05. **j** Schematic diagram of reporters of transient dual-luciferase assay. **k** Transient dual-luciferase assay in *N. benthamiana*. LUC: Firefly Luciferase; REN: Renilla Luciferase; NOS: NOS terminator. Data are mean ± SD from independent biological replicates (n = 3). The letters 'a' and 'b' indicate statistically significant differences by one-way ANOVA Duncan's test (p = 0.001926 and 0.006574 < 0.05). Source data are provided as a Source Data file.

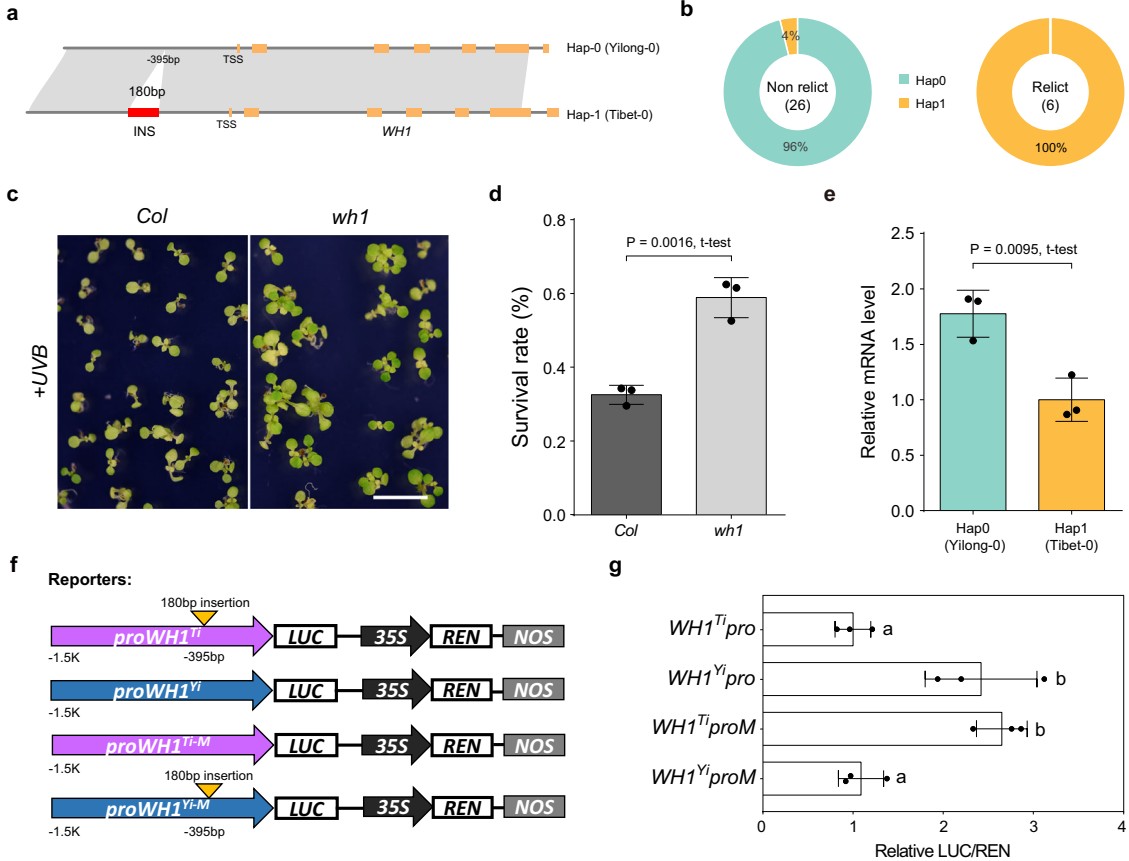

**Fig. 4 | A 180 bp insertion of the WH1 promoter in Hap1 contributes to its low transcriptional expression and resistance to UVB stress. a** Two haplotypes of *WH1* are determined by the presence or absence of 180 bp insertion (red bar) in the promoter region. TSS: transcription start site. **b** The distributions of the two haplotypes of *WH1* in relict and non-relict ecotypes. **c** WH1 negatively regulates UVB resistance. **d** Survival rates were collected after 4 days recovery. Data are mean ± SD from independent biological replicates (n = 3), and two tailed t-test was used for significance statistics. **e** The relative mRNA level of *WH1* in two haplotypes. Data are mean ± SD from independent biological replicates (n = 3), and two tailed t-test was

used for significance statistics. **f** Schematic diagram of reporters of transient dual-luciferase assay. LUC: Firefly Luciferase; REN: Renilla Luciferase; NOS: NOS terminator. *WH1* promoter fragments (1550 bp) were cloned from Yilong-0 (Hap0) or Tibet-0 (Hap1) genomic DNA. **g** Transient dual-luciferase assay in *N. benthamiana*. Data are mean ± SD from independent biological replicates (n = 3). The letters 'a' and 'b' indicate statistically significant differences by one-way ANOVA Duncan's test (p = 0.0074, 0.0030, 0.0107, and 0.0042 < 0.05). Source data are provided as a Source Data file.

variant (Hap1), for example, the relict high-altitude Tibet-0 ecotype (Fig. 4c–g). Furthermore, we found a specific 332 bp TE insertion in the promoter region of *HPCA1* in the Tibet-0 ecotype (Fig. 5a, b and Supplementary Fig. 25). This insertion was found to be associated with increased transcriptional expression of *HPCA1*, which enhanced the resistance of this ecotype to the drought stress in the high-altitude arid habitat (Fig. 5c–g). These functional tests suggest that the biallelic SVs play an important role in the local adaptation of *A. thaliana* to different ecoregions.

## Structural variants supplement a proportion of the missing heritability and were associated with the variation of multiple adaptive traits

To evaluate the power of the graph-based pan-genome in dissecting the genetic basis of adaptive traits, we detected 67,053 SVs in 1135 ecotypes by mapping Illumina short reads to our graph pan-genome. After quality control for missing rate and minor allele frequency, 20,326 SVs in 1073 ecotypes were identified as non-randomly distributed across the five chromosomes (Fig. 6a) and were kept for

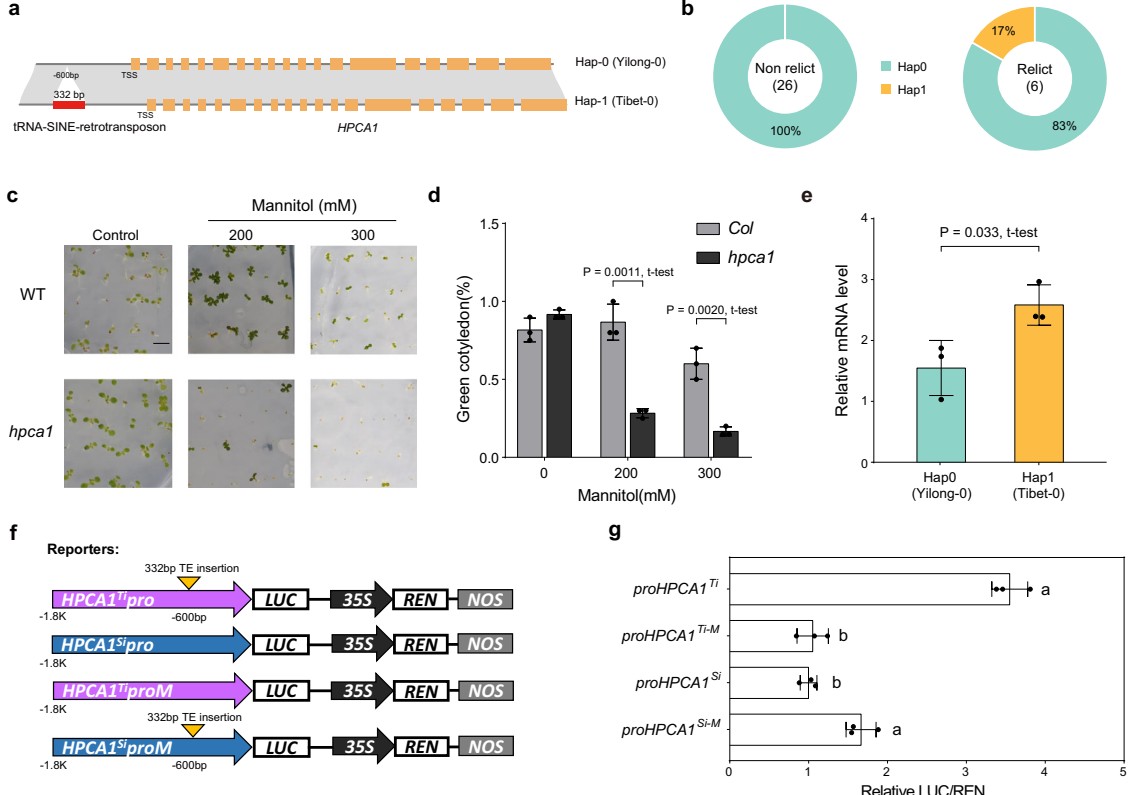

**Fig. 5 | A 332 bp transposable element (TE) insertion of the HPCA1 promoter in Hap1 contribute to its high transcriptional expression and resistance to drought stress. a** Two haplotypes of *HPCA1* are determined by the presence or absence of 332 bp TE insertion (red bar) in the promoter region. TSS: transcription start site. **b** The distributions of the two haplotypes of *HPCA1* in relict and non-relict ecotypes. **c** *HPCA1* positively regulates drought resistance. **d** Cotyledon greening rates of *Col* and *hpca1*. Results represent the mean ± SD from 3 independent experiments, and two tailed t-test was used for significance statistics. **e** The relative mRNA level of *HPCA1* in two haplotypes. Data are mean ± SD from independent biological replicates ($n = 3$), and two tailed t-test was used for significance statistics. **f** Schematic diagram of reporters of transient dual-luciferase assay. LUC: Firefly Luciferase; REN: Renilla Luciferase; NOS: NOS terminator. *HPCA1* promoter fragments (1800 bp) were cloned from Yilong-0 (Hap0) or Tibet-0 (Hap1) genomic DNA. **g** Transient dual-luciferase assay in *N. benthamiana*. Data are mean ± SD from independent biological replicates ($n = 3$). The letters 'a' and 'b' indicate statistically significant differences by one-way ANOVA Duncan's test ($p = 0.0001566, 0.01175$, and $0.0004619 < 0.05$). Source data are provided as a Source Data file.

downstream analysis. Among these SVs, only 3369 (16.57%) were tagged by SNPs (linkage disequilibrium, LD > 0.6, Fig. 6b). To evaluate the role of SVs in the variation of adaptive traits, we estimated their contribution to the variation of 61 traits, including 21 environmental variables (19 BIOCLIM, global UV-B radiation data[34] and SRTM elevation data from WorldClim v2.1[27]) in their natural habitat, as well as two flowering time measurements taken at 10 °C and 16 °C[8], and 38 ionomics phenotypes[35]. SVs were found to explain a larger proportion of phenotypic variance for 48 (78.69%) of the analyzed traits, explaining a mean of 57.98% of the phenotypic variations (Fig. 6c and Supplementary Data 1). This is 1.18% more than the proportion of variation explained by SNPs and 0.26% less than that what is explained jointly by SVs and SNPs, indicating that SVs are an important contributor of variation in adaptive traits.

Out of the 61 analyzed variables, flowing time measured at 10 °C and 11 ionomics phenotypes showed significant associations in SV-GWAS analyzes that were not detected in SNP-GWAS analysis (Supplementary Figs. 26–36 and Supplementary Data 2). For example, two SV peaks, one at chromosome 1:4,137,790 bp and a second one at chromosome 5:8,021,689 bp, were associated with the variation of flowering time measured at 10 °C (Fig. 6d). The first SV was a 77 (+/+)/85 (-/-) bp divergent sequence, where the +/+ genotype increased the flowering time by $3.17 \pm 0.53$ days ($p = 2.17 \times 10^{-11}$). The second SV peak was a 7190 bp insertion, where the +/+ genotype decreased the flowering time by $2.75 \pm 0.52$ days ($p = 8.37 \times 10^{-7}$) (Fig. 6e). No association signals were present in SNP-GWAS around this

SV, though this may be due to a low LD with surrounding SNPs (Fig. 6d). Taking these results together, the high proportion of variance explained by SVs and the detection of SV associations with environmental conditions highlighted the value of SV in determining the genetic basis of adaptive trait evolution.

## Discussion
In this study, we assembled high-quality genome sequences of 32 ecotypes in *A. thaliana*. Our phylogenomic analyzes of these ecotypes supported the previous hypothesis that *A. thaliana* experienced a postglacial expansion that produced many humid ecotypes across Eurasia and North America[7,23]. These ecotypes comprise a monophyletic lineage despite their widespread distributions. However, six paraphyletic, disjunct relict ecotypes were also analyzed, occurring in Europe, Africa, and Asia. Interestingly, the Tibet-0 ecotype was inferred to be the earliest-diverged and a sister to the other ecotypes (Fig. 1b). This phylogenomic and phylogeographic pattern suggests that *A. thaliana* may have expanded its distribution from Europe at least twice. The first expansion may have extended to the Qinghai-Tibet Plateau, where a relict ecotype is retained to the present day. Because of the strong selection pressures from this harsh alpine environment, this ecotype may have accumulated many specific mutations that caused it to be clustered as the earliest divergent ecotype in this analysis.

In addition to the 68.8 % of the pan-gene-families identified as the core families (21,575 gene families) shared by all ecotypes, the

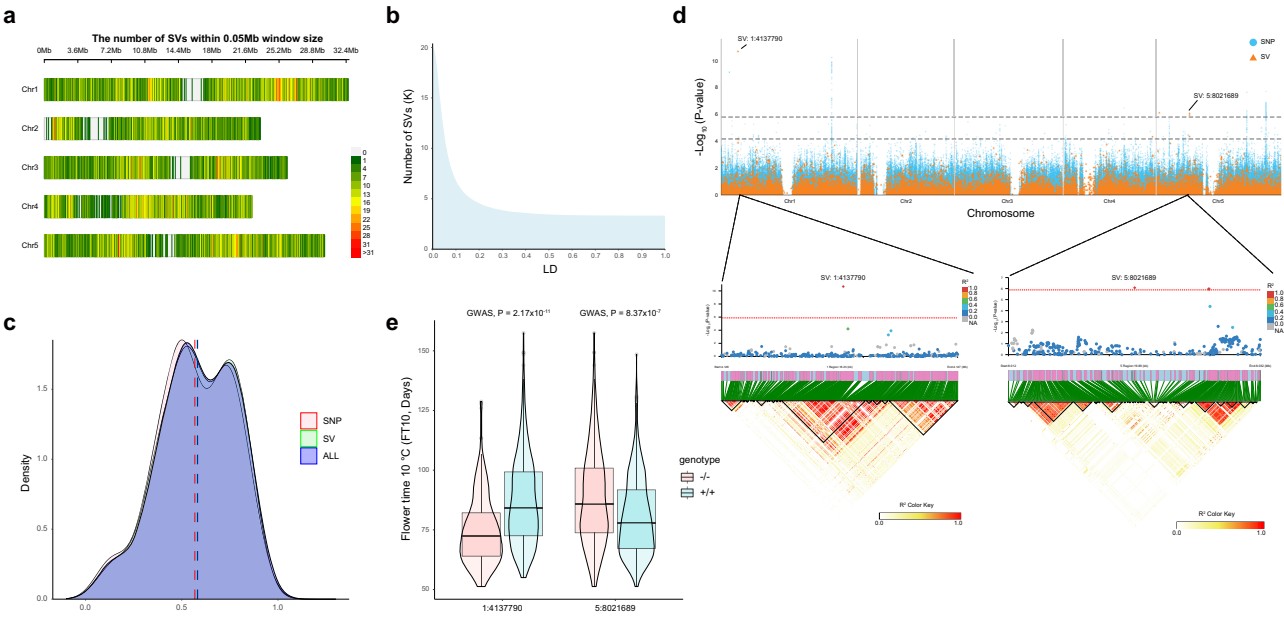

**Fig. 6 | Contribution of structural variants (SVs) to environmental adaptation.**
**a** Genomic distribution of SV from a population of 1,071 worldwide *A. thaliana* accessions from the 1001 Genomes Project (https://1001genomes.org/) and two additional ecotypes, Tibet-0 and Yilong-0. **b** Number of SVs (y-axis) tagged by SNPs at a different linkage disequilibrium (LD) cut-off (x-axis). **c** Distribution of the proportion of variance (PVE) explained by SNP, SV, and all variants (SV + SNP). **d** Top: Manhattan plot of SV-GWAS (orange) and SNP-GWAS (blue) for flowering time measured at 10 °C under greenhouse conditions. The dashed black lines are genome-wide significance thresholds for SNP-GWAS (upper, 5.80) and SV-GWAS (lower, 4.17). Middle: Zoomed in genomic regions where SV-GWAS detects unique

associations, SV Chr1:4,137,790 and SV Chr5:8,021,689. The diamonds represent the leading variants, and the colors of surrounding variants were highlighted using their LD with corresponding leading variants. Bottom: LD heatmaps of the associated regions. Significance was tested by a standard linear mixed model. **e** Boxplot illustrating the genotype and phenotype map at two SVs associated with flower time (FT) (SV:1:4137790 -/-: n = 256; +/+: n = 630; SV:5:8021689 -/-: n = 327; +/+: n = 559;) The upper and lower edges of the boxes represent the 75% and 25% quartiles, the central line denotes the median, and the whiskers extend to 1.5× the inter-quartile range (IQR). Significance was tested by a standard linear mixed model. Source data are provided as a Source Data file.

remaining 9773 gene families (the softcore, dispensable, and private types) vary greatly between ecotypes (Fig. 1d). These variable gene families are functionally enriched in stress responses and associated with climate variables. These findings suggest that gene repertoire varies greatly between ecotypes and gene birth and loss in each eco-type provide a likely basis for local adaptation. In addition, the core genes have lower Ka/Ks ratios than the variable genes across ecotypes (Fig. 1h) and tend to evolve under strong purifying selection[13,36].

A total of 61,322 SVs that overlap with 18,883 genes were identified to vary between ecotypes (Fig. 3b and Supplementary Table 13). These SVs may affect expression levels of the overlapped genes (Figs. 3e, i and 4e, l). It should be noted that more than 50% of the identified large SVs (> 500 bp) arise from the inserted TEs. Therefore, it is highly likely that jumping TEs initially created variable SVs that removed essential parts of genes, causing a reduction of function that resulted in the polymorphic repertoire and variable gene number between ecotypes. These genetic changes likely played an important role in the under-lying local adaptation of *A. thaliana* to varied habitats.

Using SVs called from 1135 re-sequenced ecotypes from the *A. thaliana* 1001 Genome Project[8] and two additional ecotypes, Tibet-0 and Yilong-0, we compared the amount of phenotypic variance explained by SVs and SNPs and found that SVs are an important source of phenotypic variation in addition to SNPs[37]. SVs supplement a pro-portion of heritability and are associated with the variation in multiple adaptive traits, highlighting their potential contribution to missing heritability and local adaptation[16]. Our assembled genomes, gene annotations, and SVs thus provide valuable resources for system-atically exploring the genetic basis underlying how SVs and the dele-tion and insertion of entire genes contribute to variation in ecological phenotypes and ecological adaptation.

## Methods
### Sample selection and sequencing
We selected 32 representative ecotypes of *A. thaliana* distributed throughout different continents, including 6 relict ecotypes, 20 of which had publicly available genome resequencing data from the *A. thaliana* 1001 Genome Project[8] (Supplementary Table 1). Seeds of the 32 ecotypes were sowed in a greenhouse at Sichuan University until the seeds germinated. Then, fresh leaves were collected and stored at −80 °C to construct HiFi SMRTbell libraries. The 15 Kb libraries were pre-pared using the SMRTbell Express Template Prep Kit 2.0 (Pacific Biosciences, CA, USA) following the manufacturer's instructions and sequenced on the PacBio Sequel II platform (Pacific Biosciences, Menlo Park, CA, USA). We used the PacBio SMRT-Analysis package (https://www.pacb.com) for quality control of the raw polymerase reads and generated the HiFi reads using SMRTLink 9.0 software with para-meters --min-passes=3 --min-rq=0.99. The final yield HiFi data of 32 ecotypes ranged from 2.18 Gb to 8.28 Gb, with coverage of around 15 to 60 X of the *A. thaliana* genome (Supplementary Table 2) based on the k-mer estimate of Col-0 genome size 137.70 Mb as reference (Supplementary Fig. 1 and Supplementary Table 5).

The total RNA of 11 *A. thaliana* ecotypes were extracted from the leaf tissues for the library construction. These libraries were subse-quently sequenced on the Illumina HiSeq X Ten platform, which pro-duced around 6 Gb of data for each sample (Supplementary Table 3). For whole genome resequencing of Tibet-0 and Yilong-0, paired-end libraries were also constructed and sequenced on the Illumina HiSeq X Ten platform (Supplementary Table 3). RNA-seq data of the other 26 ecotypes were downloaded from the NCBI SRA database under Bio-Project PRJNA187928[38], PRJEB15161, and PRJNA319904[39] (Supplemen-tary Table 4).

## De novo genome assembly of 32 ecotypes

The genome size, heterozygosity, and repeat ratio of the reference Col-0 genome were estimated based on a 17-bp k-mer frequency analysis by GenomeScope v2.0[40] with parameter '-k 17' and Jellyfish v2.2.9[41] with parameter '-m 17 --min-quality=20 --quality-start=33' using NGS data download from CRA004538[5] in CNCB database. Genomes of the 32 sequenced ecotypes were assembled by hifiasm v 0.18[42] using CCS reads, with parameters '-l0' to disable duplication purging, which may introduce misassemblies if a species has low heterozygosity. There are two outputs of raw hifiasm assemblies: the primary assembly (p_ctg) and the alternate assembly (a_ctg), we selected p_ctg for further assembly and downstream analyzes. In order to construct 5 pseudo-chromosomes of each *A. thaliana* ecotype, we used RagTag v 2.1.0[43] to scaffold the contigs based on the recently published telomere-to-telomere (T2T) genome assembly Col-PEK[6]. The completeness of each assembly was estimated using the embryophyta_odb10 database by Benchmarking Universal Single-Copy Orthologs (BUSCO) v.5.0.2[24] with default parameters.

## Identification and annotation of repetitive elements

To structurally annotate transposable elements (TEs) in the 32 assembled genomes, we used the Extensive De-Novo TE Annotator (EDTA) v.2.1.0[44] with parameter '--species others --sensitive 1 --step all --anno 1 --u 7e-9' to generate the non-redundant de novo TE libraries and annotated the intact long terminal repeat retrotransposons (LTRs) for each ecotype. The insertion time of each intact LTR was also provided by the software. The generated TE libraries and *Arabidopsis* repeats in RepBase were further passed into pan-EDTA[45] to generate the pan-TE library. Repeat regions of the 32 genomes were then re-masked by RepeatMasker v 4.1.2-p1[46] with default parameters using the pan-TE library. For overlapping repeats, the overlapped regions were split in the middle. To estimate the repetitive elements continuity of each assembly, the LTR assembly index (LAI) was calculated by LTR_retriever v 2.8[47] using intact LTR datasets.

## Prediction of protein-coding genes

In order to obtain high-quality gene structure annotation of each ecotype, we combined three methods: ab initio, protein homology, and transcriptome-based annotation. Firstly, we aligned RNA-seq reads to each genome using HISAT2 v 2.1.1[48] with parameter '--dta' and assembled transcripts using StringTie v 2.1.4[49] with parameter '--rf'. The assembled transcripts were then passed to PASA v.2.3.3[50] after filtering by seqclean to generate Open Reading Frames (ORFs). The predicted complete, multi-exon genes models then had redundant high identity removed (with an all-to-all identity cut off of 70%) and were subsequently and sent to train the Hidden Markov Model for AUGUSTUS v 3.2.3[51]. In order to further support gene annotation by AUGUSTUS, we also used bam2hints from AUGUSTUS to generate an intron hints file based on a bam file generated by HISAT2. We used this hints file to carry out ab initio gene prediction by AUGUSTUS using default parameters. For homologous protein prediction, protein sequences of Araport11[25] were downloaded from TAIR (https://www.arabidopsis.org/) and aligned against each genome using TBLASTN[52] with parameters '-e 1e-5'. After filtering low-quality results, the gene structure was predicted using GeneWise v 2.4.1[53]. The results of PASA, AUGUSTUS, and GeneWise were combined using EvidenceModeler v 1.1.1[50] to generate a combined protein-coding gene set. After merging, we filtered out incomplete gene models and gene models overlapping with repeats if the overlap ratio of CDS region were more than 80%. For genes with CDS lengths less than 150 bp or less than 750 bp and 3 CDS, we used the Pfam database for validation. If no alignment result was obtained or the alignment coverage was less than 25%, the gene model was filtered out.

As for the model plant, gene numbers starting with ATXG are widely used in *A. thaliana*. In order to minimize the difference from previous gene annotations, we use Liftoff v 1.6.3[54] to map the Araport11 gene annotation onto each genome with parameter '-exclude_partial -a 0.9 -s 0.9 -polish' and replaced our gene annotation which overlaps with the Araport11 gene (valid_ORF=True). The final gene set was named such as col_AT1G01010 (mapped by Araport11) and col00072 (unmapped or newly annotated). The longest transcript of each predicated gene model was considered as the representative for further analysis. The completeness of gene annotations was also estimated by BUSCO using the embryophyta_odb10 database with default parameters.

For gene functional annotation, eggNOG-mapper v2[26] was applied to obtain seed ortholog and functional description, Gene Ontology (GO) numbers, Enzyme Commission nomenclature (EC) numbers, Kyoto Encyclopedia of Genes and Genomes (KEGG) numbers, PFAM numbers and so on.

## Phylogenetic analysis

In order to construct phylogenetic relationships among 32 ecotypes of *A. thaliana*, protein sequences from *A. lyrata* were downloaded from Phytozome v13[55] and used as an outgroup. Then we did an all-to-all blastp with peptide sequences of protein-coding genes annotated from these 33 genomes by NCBI BLAST v 2.2.30 + [52] with cut-off e-values of 1e-5 and then input the results into OrthoFinder[56] for gene clustering with parameter '-I 1.5'. The single-copy orthologous genes were further extracted from OrthoFinder results, protein sequences were aligned by MAFFT v 7.490[57] and conserved sites from multiple sequence alignment were extracted by Gblocks v 0.91b[58]. The phylogenetic tree was constructed by IQ-TREE v 2.0.3[59] with parameter '-m MFP -B 1000 --bnni' to automatically find the best model and perform 1000 ultrafast bootstrap analyzes to test the robustness of each branch.

## Construction of the protein-coding gene-based pan-genome

We did an all-to-all blastp with protein sequences of the 32 *A. thaliana* ecotypes with parameter '-e 1e-5' and input the result file into OrthoFinder for gene family construction by setting the inflation factor to 1.5. Finally, we obtained 31,317 non-redundant gene clusters. We then classified those clusters into 4 categories: core gene clusters that were conserved in all 32 ecotypes; soft-core gene clusters, which were present in 26–31 ecotypes; dispensable gene clusters, which were found in 2–25 genomes; and private gene clusters, which contained genes from only 1 sample (including unassigned genes). The longest encoded protein was chosen to represent each gene. In order to further simulate the number of protein-coding genes in the pan-genome and core genome, we used PanGP v 1.0.1[60] with a completely random algorithm setting sample size to 1000 and sample repeat to 30 based on the OrthoFinder results.

## Identification of environment-associated variable gene families

Environmental data from 1970 to 2000 for the 19 BIOCLIM variables was downloaded from WorldClim v2.1 (www.worldclim.org)[27] with a spatial resolution of 30 seconds (~1 km²). Principal component analysis (PCA) of 32 *A. thaliana* ecotypes based on variable gene families was performed by function rda() in the R package vegan[61]. Multiple regression of 19 BIOCLIM variables on selected ordination axes was performed by function env.fit() in the R package vegan with significance determined using 99,999 permutations. Variable gene families which were significantly associated with BIOCLIM variables were further identified by logistic modeling using the glm() function with parameter 'family = "binomial"'.

## Gene expression analysis

We first removed the adaptor sequences and discarded the low-quality reads using Trimmomatic v 0.38[62] with parameter values 'SE ILLUMINACLIP:TruSeq3-SE.fa:2:30:10 LEADING:3 TRAILING:3

SLIDINGWINDOW:4:15 MINLEN:36 TOPHRED33' for single-end RNA-seq reads and 'PE ILLUMINACLIP:TruSeq3-PE.fa:2:30:10 LEADING:3 TRAILING:3 SLIDINGWINDOW:4:15 MINLEN:36 TOPHRED33' for paired-end RNA-seq reads. Then the clean reads were mapped to the reference genomes using HISAT2. The expression levels of each gene were calculated in FPKM (fragments per kilobase of exon model per million mapped fragments) using StringTie with the default parameters.

### Ka/Ks calculation of different types of pan-genes
Non-synonymous substitution rates (Ka), synonymous substitution rates (Ks), and Ka/Ks in core, softcore, and dispensable gene clusters were computed using the KaKs_Calculator v 2.0[63] with default parameters. We conducted amino acid alignment for gene pairs in each cluster first and then converted the results into the coding sequence (CDS) alignment using PAL2NAL v 14[64]. The alignments were further passed to the KaKs_Calculator to obtain Ka/Ks values.

### Construction of the graph-based pan-genome and SVs calling
We used minigraph[65] to construct the graph pangenome of the 32 high-quality *A. thaliana* genome assemblies based on sequence alignment using a modified minimap2 with the parameter '-cxggs'. The Col-0 genome assembled in this study was set as the reference and the other 31 genomes were added into the multi-assembly graph successively. The fragments differing from the reference genome are displayed as different paths in the generated graphical fragment assembly (GFA) file. If two or more paths are connected between two fragments, they will form bubbles.

The minigraph graph consists of chains of bubbles with the reference sequences as the backbone. Each bubble in the graph represents a structural variation. In order to call structural variations based on bubbles, we used gfatools (https://github.com/lh3/gfatools) to get the position of each variation. Extracted structural variations were further classified into biallelic (two paths in a bubble) and multiallelic (more than two paths in a bubble) types.

To genotype the SVs in the 1135 *A. thaliana* individuals downloaded from the NCBI SRA database under BioProject PRJNA273563. Tibet-0 and Yilong-0 were sequenced in this study. We mapped the short reads from each individual to the graph-based pan-genome via vg toolkit v1.40.0-88-g04775076b[20] using default parameters. After filtering individuals with a missing rate above 0.5 or minor allele frequency (MAF) above 0.05 using Plink v1.90b6.7[66], a total of 1,073 individuals with 20,326 SVs were passed. Then these SVs were imputed using beagle.22Jul22.46e[67].

### Structural variation gene identification and verification
In order to obtain the actual chromosome position and gene region overlap of SV in different ecotypes, we conducted a whole-genome alignment of 32 *A. thaliana* genomes. The 32 HiFi assembled genomes were aligned to the Col-0 reference genome using Minimap2 v.2.16[68] with default parameters; alignments lengths shorter than 1000 bp were discarded. The results show the real positions of each graph pangenome segment in the different genomes and, in combination with the gene annotation files, identify the SV genes in the different ecotypes. In order to verify the corresponding relationship between SV genes, we used MCScanX[69] to perform gene collinearity analysis with default parameters.

In order to confirm the SV genes, we mapped HiFi reads to the genome using minimap2 to eliminate assembly errors, while the RNA-seq reads were mapped to the genome using HISAT2 to rule out incorrect gene structure annotations.

### SNP calling
For SNP database construction, the resequencing reads of the 1135 individuals as well as Tibet-0 and Yilong-0 sequencing data were mapped to the Col-0 reference genome in this study with the bwa-mem2 algorithm of BWA v0.7.17-r1188[70] using default parameters. The resulting BAM files were further filtered using SAMtools v1.3.1[71] for non-unique and unmapped reads and Picard tools v1.87 (http://broadinstitute.github.io/picard/) for duplication. SNP calling was carried out using the Genome Analysis Toolkit (GATK) v4.2[72] with default parameters. After filtering via plink with parameters '--geno 0.1 --maf 0.03 --mind 0.1', a total of 2,033,562 SNPs were retained for downstream analysis.

### Genome-wide association analysis for 61 traits
For each ecotypes, 21 environmental variables (19 BIOCLIM global UV-B radiation data (https://www.ufz.de/gluv) and SRTM elevation data from WorldClim v2.1 (www.worldclim.org)), two flowering time traits measured at 10 °C and 16 °C, and 38 ionomics phenotypes (https://ffionexplorer.nottingham.ac.uk/ionmap) were used to evaluate the role of SVs in dissecting the genetic basis of adaptive traits. Downloaded phenotypes data were standardized before being subjected to downstream analysis. We used the standard linear mixed model implemented in GCTA[73] to perform a genome-wide association analysis for SVs and SNPs. The genetic variants were first filtered by removing alleles with a frequency less than 0.05. A kinship was calculated with the genome-wide marker and was used to account for confounding with population structure.

### Partitioning the phenotypic variance to SVs and SNPs
We used the following mixed linear model (**1**) to partition the phenotypic variance to SVs and SNPs.

$$Y = \mu + Zu + e \qquad (1)$$

Y is a vector of phenotype, e is the normally distributed residual. $\mu$ is the population mean, and u is a random effect vector of polygenic scores. Z is the corresponding design matrix obtained from a Cholesky decomposition of the kinship matrix G, estimated using the genome-wide markers, excluding the detected QTLs using GCTA[73]. The Z matrix satisfies ZZ' = G, therefore, u ~ $N(0, I\sigma_g^2)$. We derived the kinship matrix G from SV and SNP individually and estimated their heritability by fitting a linear mixed model with the corresponding kinship as a covariance structure implemented in the R package hglm[74]. Variance explained by kinship is calculated as the interclass correlation (**2**).

$$\text{Variance explained by kinship} = \frac{\sigma_g^2}{\sigma_g^2 + \sigma_e^2} \qquad (2)$$

In order to estimate the joint contribution from SVs and SNPs, a composite model with two random effects was fitted (**3**).

$$Y = \mu + Z_1 u_1 + Z_2 u_2 + e \qquad (3)$$

Y, $\mu$, and e is the same as described in **1**. $u_1$ is a random effect vector aggregating the effects from all the SNPs while $u_2$ is a random effect vector aggregating the effects from all the SVs. $Z_1$ and $Z_2$ is the corresponding design matrix obtained by decomposing the corresponding kinship matrix estimated from SNPs and SVs as described above. Then, the proportion of variance explained by SNPs and SVs were estimated as below:

$$\text{Proportion of variance explained by SNPs} = \frac{\sigma_{u1}^2}{\sigma_{u1}^2 + \sigma_{u2}^2 + \sigma_e^2} \qquad (4)$$

$$\text{Proportion of variance explained by SVs} = \frac{\sigma_{u2}^2}{\sigma_{u1}^2 + \sigma_{u2}^2 + \sigma_e^2} \qquad (5)$$

### RT-qPCR for the *WH1*, *HPCA1*, *CCR1* and *KNAT3* genes

Total RNA was isolated using the TRIzol method from Tibet-0 and Yilong-0 seedlings. Quality and integrity of the extracted RNA were determined using a NanoDrop 2000 spectrophotometer (Thermo Scientific, Waltham, MA, USA) and 2% agarose gel electrophoresis. We then used the Hifair®III 1st Strand cDNA Synthesis Kit (Yeasen Biotech Co., Ltd, Shanghai, China) to reverse-transcribe the quantified RNA into cDNA. Quantitative Real-time PCR of SV genes was then performed with a Bio-Rad CFX384 Real-Time PCR Detection System (Bio-Rad, USA) using Hifair UNICON Universal Blue qPCR SYBR Green Master Mix (Yeasen Biotech Co., Ltd, Shanghai, China) and the primer sets. Each experiment was independently performed three times. Data were normalized to *EIF4A* by $2^{-\Delta\Delta CT}$ analysis. The primer sequences used for qRT-PCR analysis are shown in Supplementary Table 16.

### In vivo dual-luciferase activity assays

In vivo dual-luciferase activity assays were carried out using tender *Nicotiana benthamiana* leaves and the pGreen II 0800-LUC vector system[29]. Agrobacterium tumefaciens GV3101 strains harboring the promoter variants of *WH1* (*WH1^Ti^pro*, *WH1^Yi^pro*, *WH1^Ti^proM*, *WH1^Yi^proM*), *KNAT3* (*proKNAT3^Ti^*, *proKNAT3^Yi^*, *proKNAT3^Ti-M^*) or *CCR1* (*mini35S:LUC, SV(TE)+mini35S:LUC*) were each infiltrated using a syringe into separate *N. benthamiana* leaves at an OD600 = 0.6. The infiltrated plants were kept in the dark for 2 days and then 1 day under normal conditions, after which measurements of Firefly Luciferase (LUC) and Renilla Luciferase (REN) contents were taken using a Dual-Luciferase® Reporter Assay System kit (Promega, Madison, WI, USA) according to the manufacturer's instructions. Three independent experiments were performed. One-way ANOVA multiple comparisons (Turkey's multiple comparison test) or unpaired *t* test was used in the statistical analysis. The primer sequences used are shown in Supplementary Data 3.

### Reporting summary

Further information on research design is available in the Nature Portfolio Reporting Summary linked to this article.

## Data availability

The raw sequencing data for the PacBio HiFi reads, RNA sequencing reads, and resequencing Illumina short reads have been deposited in the Genome Sequence Archive (GSA)[75] database at the National Genomics Data Center, Beijing Institute of Genomics, Chinese Academy of Sciences/China National Center for Bioinformation under BioProject PRJCA012695. The genome assembly, genome annotation, pan-TE library, graph pan-genome, gene family and gene presence/absence matrices files have been deposited in Figshare [https://figshare.com/articles/dataset/32_ecotypes_Arabidopsis_thaliana_genomes_gene_annotation_pan-TE_library_graph_pan-genome_gene_family_and_gene_presence_absence_matrices_files_/21673895]. Public RNA-seq data were downloaded from the NCBI SRA database under BioProject PRJNA187928, PRJEB15161, and PRJNA319904. The resequencing data of a total of 1135 individuals were downloaded from PRJNA273563. The 19 BIOCLIM and SRTM elevation data used in this study were download from WorldClim v2.1 (www.worldclim.org). The global UV-B radiation data was download from gIUV (https://www.ufz.de/gluv). Source data are provided with this paper.

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

## Acknowledgements

The work was supported by the Natural Science Foundation of China (32030006), the Second Tibetan Plateau Scientific Expedition and Research (STEP) program (2019QZKK0502), the Strategic Priority Research Program of Chinese Academy of Sciences (XDB31000000), and the Talent Introduction Research Start-up Fund by Lanzhou University (561120221). Thanks to the support of the Supercomputer Center of Lanzhou University, especially to the arm cluster of the Center for providing computing resources in structural variation prediction. We are grateful for Genesis Technology Communication (Beijing) Co. Ltd for English improvements.

## Author contributions

J. L. led the research. S. L. and Yanjun Z. co-directed the program. S.L. prepared all materials. M.K., H.W., Wenyu L., M.Z., Y.H., Wei L., and C.C. performed the bioinformatics analysis. H.L., Y.S., L.T., K.Y., Yusen Z., and Z.Y. designed and performed functional experiments. M.K. and J.L. wrote the manuscript. M.K., S.L., and Yanjun Z. revised the manuscript. All authors discussed the results and commented on the manuscript.

## Competing interests

The authors declare no competing interests.
