## [Peer Review File · Nature Communications]

The pan-genome and local adaptation of *Arabidopsis thaliana*Reviewers' Comments:

Reviewer #1:

Remarks to the Author:

In this study, the authors constructed a normal gene-based pan-genome and a graphed pan-genome for *Arabidopsis thaliana* with newly assembled 38 genomes including relict ecotypes. With such pan-genomes, they analyzed the variations of gene, TE and sequences across them. They focused on the SVs, and identified some potential SV of genes for the local adaptation of different ecotypes such as one high-altitude relict ecotype in Tibet. Overall, the manuscript is well organized in general. Such genome resources can also contribute the research community of *Arabidopsis*. Here I have some comments regarding the data analyses of SV.

1. The authors used a graph-based SV tools to detect SV. However, some types of SV like inversion, translocation and duplication were not found in their results. Compared to another study (Jiao et al., Nat. Commun., 2020) with 8 *thaliana* genomes, the SV detected in this study seems much fewer. Besides, the SV density gets much lower in centromeres as shown in Figure 3f, which is not consistent with the previous study (Jiao et al., Nat. Commun., 2020). A combination of graph-based and assembly mapping-based should provide a more comprehensive repertoire of SVs.
2. Line 238-240, only 20K SVs were genotyped with the graph-based pan-genome. So how are other 42k SVs? Why are they not genotyped?
3. The pan-genome plot in fig 3a is apparently different with the fig 1c. The pan-genome size still goes up when 29 or more genomes are included, which is not as the line 185-186 says. Besides, the fitting curve is missing in fig 3a.

Minor comments:

1. Line 47-49, some previous studies with multiple de novo *Arabidopsis* genomes can be cited such as Jiao et al., Nat. Commun., 2020
2. Line 229-231, this insertion may reduce the expression of *KNAT3*, but it is not necessary to say "Because of this insertion" because a lot of other sequence variations exist between Tibet-0 and other ecotypes.
3. Fig 2i, fig 3d and h, the number in color of white are not clear in printout.

Reviewer #2:

Remarks to the Author:

In this manuscript, the authors constructed the pan-genome of the model plant *A. thaliana*. To this end, they applied standard gene-based and graph-based approaches to 38 newly-assembled genomes. In addition, the authors constructed a pan-collection of transposable elements and demonstrated the use of structural variants analysis in the context of GWAS of several ecological phenotypes related to local adaptation. While I find this manuscript to be limited in terms of novelty and biological insights, I still believe the data produced is a valuable resource for future research. However, the authors do not provide any means for convenient data access and analysis. In addition, for some of the accessions, the coverage was rather low (15x). While the BUSCO scores were still sufficient for gene-based discovery, I am not sure this is so regarding transposable elements and SV detection. Lastly, the writing requires some improvement in certain sections (see below).

Major

1. Data availability and accessibility - to make the *A. thaliana* pan-genome a useful resource for the community, some means for data access and browsing should be developed. As far as I can tell, currently only the raw data, assemblies and annotations can be downloaded. It would be beneficial to

make available also some of the additional analyses, e.g. TE libraries, gene families, the graph reference, gene presence/absence matrices etc. Ideally, the data should be made available through some web interface or integrated into a larger project (1001 genomes?), but at the very least all the data should be made available.

2. There is at least one previous publication where high-quality assemblies were generated (Jiao & Schneeberger, 2019). There may be more (see the data center in the 1001 genomes website). I suggest adding these to the pan-genome. In that case, I would advise on annotating the assemblies using your pipeline to ensure results consistency.

3. The genomes were assembled into contigs (N50 5-20 Mb) and then scaffolded based on the reference genome. These assemblies were then used for the graph construction and SV calling. I suspect this procedure may result in overlooking some large-scale SVs relative to the reference.

4. In the annotation pipeline, it seems like no filtration of the predicted genes was performed. Is that so? For instance, do the annotations include gene models derived solely from ab-initio prediction? In my experience, tools like AUGUSTUS tend to over-annotate, so some filtration or QC of the annotations could be useful. One option would be using Annotation Edit Distance (AED) cutoff.

5. I am a bit skeptical of the phylogenetic analysis presented here. It is often difficult to infer a reliable phylogeny within a species due to incomplete lineage sorting and hybridization. That might be the reason for the relatively low bootstrap values. In addition, the inference was based on concatenation of a large number of genes, so a partitioned analysis might be required. It might be a good idea to:
a) Reconstruct the phylogeny using SNP data and see if the resulting topologies are similar.
b) Use a dimensionality reduction method like PCA to visualize the distances between genomes and see if the conclusions are the same.

6. Regarding the analysis of the pan-TE: Can the authors relate this to the coding gene classification? E.g., do core genes tend to have less/more TEs etc.

7. Regarding the SV analysis, it would be interesting to see if SVs can explain gene presence/absence patterns obtained in the annotation-based pan-genome.

8. As written above, it is preferential to have higher coverage for some accessions.

9. The authors claim that some of the SV identified are the cause for some detected function or changes in expression (e.g., line 228-229), but this is more of a speculation, without any functional validation – can the authors functionally demonstrate that a given SV leads to differences in expression or function?

10.

Minor

As written above, the writing can be improved. Here are some examples.

* Lines 6-7: better explain the limitations of a single reference genome.

* Line 7: "chromosome-level" instead of "chromosomal"

* Line 8: I'm not sure all readers will be familiar with the term "relict ecotype" – can briefly explain. Also "including six relict..." would be better.

* Line 9: "several thousand" - how many exactly?

* Lines 11-12: "this species substantially expands its gene repertoire for local adaptation" - unclear, please rephrase.

* Line 15: "...was found to be specific...".

* Line 18: "...captured much of the missing heritability".

* Line 25: "studies" instead of "research".

* Line 27: "complexity" - do you mean diversity?

* Line 40: is it really unknown if genetic variation contributes to local adaptation? I suspect this had been explored before.

* Line 54 - please expand a bit more on relict ecotypes and better explain why you decided to include these.

* Line 70: "Chromosome-level genome assemblies and annotation of 38 ecotypes" (or break into two sections).

- * Line 82: "contiguity" instead of "continuity".
- * Line 85: "liftoff mapping" - not clear to everyone. Maybe use the general term "gene leftover" and explain the idea.
- * Lines 87-90: the rationale and procedure are not clear. Please rewrite.
- * Line 100: I suggest "... to construct a maximum likelihood (ML) phylogeny."
- * Line 134. Change to: "...variable genes evolve more rapidly and may obtain new functions or more rapidly adapt to the environment."
- * I didn't follow the difference between the content written in line 143 and then 147-148.
- *Line 154 – It is unclear how exactly genes with TE insertion were defined
- * Line 181: "The specific sequences in each ecotype varied from 56.58 Kb to 8.45 Mb" - unclear. Do you mean novel sequences?
- * I suspect that most potential readers are not familiar with the characteristics of the pan-graph. Few sentences should be added as a preface to that section (line 176 onward) – what are the meanings of the nodes and the edges. More so, can we learn anything from the reported statistics regarding the graph's properties? What can we learn from the fact that the graph has 457k nodes? Is this a lot? Does it provide any information? How is this compared to other species?
- *Line 186 Here and in line 108 – how exactly the plateau point was identified? By eye or using any statistical measure?
- *Line 194-5. Similarly, not much insights we can gain from the statistics given here – can these be compared to other species?
- *Line 238-246. What does it mean "missing heritability? What does it mean "SVs were found to contribute more shares of heritability"?
- * Line 519: the link doesn't work ("DOI not found"), so currently the data cannot be accessed.
- * Line 688-9: "Branches with 100 bootstrap values are not shown" - do you mean less than 100? but the branches in figure 1b show lower numbers. It is written in the methods that 1000 BS were used, so I'm confused. Also "green branches" - it looks blue to me.

Reviewer #3:

Remarks to the Author:

The authors sequenced and assembled the genomes of 38 *A. thaliana* accessions. This is an impressive dataset and will likely be of considerable value to the research community, though the paper provided few novel insights.

Mostly, I was left unconvinced that data support claims regarding the adaptive value of the variation described. Certainly, SV, TE, etc contribute to adaptive differentiation across climate gradients but the results presented in the paper to explore this were not very compelling.

For CCR1 and KNAT3, the authors showed that the accession with an SV had lower expression. No experiments were conducted that confirm the causality of that SV on expression, nor on the adaptive value in relation to the environment. As such phrasing like "Because of this insertion, the gene expression level in Tibet-0 was significantly reduced compared with other ecotypes without this insertion" (line 229) are not warranted.

Fig.5d. Why is the y-axis so high?

Line 116-117: what is a "basic" biological process? All of the processes listed are quite complex. Can the authors find a more precise word?

133:134: Authors should also mention that the difference in Ka/Ks could simply be relaxed selection on non-core genes (rather than diversification due to selection for novel functions or adaptation).

The authors observed bias in the distribution of SV with relation to gene features. This is an

interesting result of the paper. However, it is unclear whether the discussion of this is implying this distribution reflect bias in the insertion site or bias in selection on "random" TE insertions.

169: Is this a mechanistic explanation or an adaptive explanation?

173:-174. Again, can't tell if this is a hypothesis about preference in insertion sites or an adaptive hypothesis.

This would be a good paper to check out <https://www.ncbi.nlm.nih.gov/pmc/articles/PMC6668482/> Shows clearly the basis of biased insertion by TE which likely is relevant for these findings.

It is very unclear what heritability is being measured. The authors talk about partitioning phenotyping variance, but it was not clear what phenotype was even measured. The methods section does not mention any traits, it only (briefly) describes the analyses.

In any case, it seems hardly accurate that the SV explained "the missing heritability" as stated in the abstract or "supplement a large proportion of heritability" (line 299). The improvement with SV was marginal: 0.03% greater than SNPS alone.

The authors should mention and cite <https://www.nature.com/articles/s41467-020-14779-y>

The link to data available <https://dx.doi.org/10.6084/m9.figshare.21673895> doesn't work. It would be helpful so we can review the quality and accessibility of the data.

The paper is written well and I think this could be a very highly cited and valuable paper given the magnitude of the data, but the follow-up analyses presented to glean some ecological or biological insights from the data do feel rather underwhelming.

REVIEWER COMMENTS

Reviewer #1 (Remarks to the Author):

In this study, the authors constructed a normal gene-based pan-genome and a graphed pan-genome for *Arabidopsis thaliana* with newly assembled 38 genomes including relict ecotypes. With such pan-genomes, they analyzed the variations of gene, TE and sequences across them. They focused on the SVs, and identified some potential SV of genes for the local adaptation of different ecotypes such as one high-altitude relict ecotype in Tibet. Overall, the manuscript is well organized in general. Such genome resources can also contribute the research community of *Arabidopsis*.

Here I have some comments regarding the data analyses of SV.

1. The authors used a graph-based SV tools to detect SV. However, some types of SV like inversion, translocation and duplication were not found in their results. Compared to another study (Jiao et al., Nat. Commun., 2020) with 8 *thaliana* genomes, the SV detected in this study seems much fewer. Besides, the SV density gets much lower in centromeres as shown in Figure 3f, which is not consistent with the previous study (Jiao et al., Nat. Commun., 2020). A combination of graph-based and assembly mapping-based should provide a more comprehensive repertoire of SVs.

Response: In order to reduce the likely errors, we removed six ecotypes that derived from the single postglacial glacial expansion with the low coverage and annotation quality. All 6 relict ecotypes have the high coverage and annotation quality. We added some description of the SVs classification in the main text: “The biallelic SVs were further divided into insertion, deletion, and divergent types according to the paths (including traditional types of SVs: inversion (divergent), translocation (one insertion and one deletion) and duplication (one insertion)) (Fig. 3c)”. As for the relatively low number of SVs and the SVs hotspots are not located in the centromeres as previously reported, we compared our newly assembled genome with T2T genome (Col-PEK) and eight previously published genomes (Fig R1). We found that the higher number of SVs and their concentration in the centromeres in the previous study may be related to the assembly quality of the centromeres. The centromeres assembled in the previously published genomes had a higher number of contigs, most of which were less than 1 Mb, indicating fragmented assembly. In addition, the structural variation dataset we used in this study only includes SVs greater than 50 bp, which may result in a lower number of SVs compared to SyRI result.

Fig R1. Comparison of genome assemblies of 41 *A. thaliana* accessions.

2. Line 238-240, only 20K SVs were genotyped with the graph-based pan-genome. So how are other 42k SVs? Why are they not genotyped?

Response: In total, we genotyped 67,053 SVs by mapping Illumina short reads to the graph-based pan-genome. Since a large proportion of these SVs were filtered out due to low minor allele frequency (5,124, Fig R2a) and high missing rate (41,603, Fig R2b) that were caused by

poor input data quality (Fig R2c/d), we only kept 20,326 SVs for downstream analysis. We have clarified this in revised manuscript.

Fig R2. Evaluation of SV calling from Illumina short reads. a. Histogram of SV MAF frequency distribution. b. Histogram of SV genotype missing rate frequency distribution. c. Fitted regression of individual sequencing data size and genotype missing rate frequency. d. Linear regression of genotype missing rate on data sequencing metric the average spot length of reads (AvgSpotLen).

3. The pan-genome plot in fig 3a is apparently different with the fig 1c. The pan-genome size still goes up when 29 or more genomes are included, which is not as the line 185-186 says. Besides, the fitting curve is missing in fig 3a.

Response: We modified this sentence into “The pan-genome size increased with the number of genomes added.” and added the fitting curve in Fig 3a.

Minor comments:

1. Line 47-49, some previous studies with multiple de novo Arabidopsis genomes can be cited such as Jiao et al., Nat. Commun., 2020

Response: We cited this reference article in the revised manuscript.

2. Line 229-231, this insertion may reduce the expression of *KNAT3*, but it is not necessary to say “Because of this insertion” because a lot of other sequence variations exist between Tibet-0 and other ecotypes.

Response: We added the in vivo dual-luciferase (Dual-LUC) activity assays of *KNAT3* gene and demonstrated the effect of the 566bp TE insertion in Tibet-0. And we modified this sentence into “The *KNAT3* gene expression level in Tibet-0 was significantly increased

compared with other ecotypes without this insertion, and in vivo dual-luciferase (Dual-LUC) activity assays demonstrated the effect of the 566bp TE insertion (Fig. 3i, j, k)."

3. Fig 2i, fig 3d and h, the number in color of white are not clear in printout.

Response: We modified the number in color of white into black and enlarged the font size.

Reviewer #2 (Remarks to the Author):

In this manuscript, the authors constructed the pan-genome of the model plant *A. thaliana*. To this end, they applied standard gene-based and graph-based approaches to 38 newly-assembled genomes. In addition, the authors constructed a pan-collection of transposable elements and demonstrated the use of structural variants analysis in the context of GWAS of several ecological phenotypes related to local adaptation. While I find this manuscript to be limited in terms of novelty and biological insights, I still believe the data produced is a valuable resource for future research. However, the authors do not provide any means for convenient data access and analysis. In addition, for some of the accessions, the coverage was rather low (15x). While the BUSCO scores were still sufficient for gene-based discovery, I am not sure this is so regarding transposable elements and SV detection. Lastly, the writing requires some improvement in certain sections (see below).

Major

1.Data availability and accessibility - to make the *A. thaliana* pan-genome a useful resource for the community, some means for data access and browsing should be developed. As far as I can tell, currently only the raw data, assemblies and annotations can be downloaded. It would be beneficial to make available also some of the additional analyses, e.g. TE libraries, gene families, the graph reference, gene presence/absence matrices etc. Ideally, the data should be made available through some web interface or integrated into a larger project (1001 genomes?), but at the very least all the data should be made available.

Response: The genome assembly, genome annotation, pan-TE library, graph pan-genome, gene family and gene presence/absence matrices files have been deposited in the Figshare database (<https://dx.doi.org/10.6084/m9.figshare.21673895>). The private link for reviewers is <https://figshare.com/s/60f2b253e4648767ddf2> .

2.There is at least one previous publication where high-quality assemblies were generated (Jiao & Schneeberger, 2019). There may be more (see the data center in the 1001 genomes website). I suggest adding these to the pan-genome. In that case, I would advise on annotating the assemblies using your pipeline to ensure results consistency.

Response: As shown in Fig R1, previously published 8 genomes by Jiao & Schneeberger had a large number of contigs assembled in the centromere region that were smaller than 1 Mb, which may lead to bias in the identification of SV hotspots and the number of SVs, thereby affecting the results of GWAS and other analyses. In addition, after the completion of the T2T genome assembly, the size of the *A. thaliana* genome expanded to ~130 Mb, while the 8 published genomes were all ~120 Mb. In order to ensure the accuracy of pan-genome analysis results, these previously assembled genome data were not included, but this article was cited

in the Introduction part. We also deleted 6 ecotypes drive from the single postglacial expansion in our study with the low coverage and annotations.

3. The genomes were assembled into contigs (N50 5-20 Mb) and then scaffolded based on the reference genome. These assemblies were then used for the graph construction and SV calling. I suspect this procedure may result in overlooking some large-scale SVs relative to the reference.

Response: Undeniably, the method of anchored contigs onto chromosomes based on homologous alignment may lose some large segments of SVs. However, due to the long read length and high accuracy of HiFi sequencing, the original contig continuity obtained by hifiasm is good (Fig R1). The 32 genome contigs assembled contain only 17-56 contigs (Supplementary Table 6). Assuming that each contig is an unidentified large segment SV, a total of 1092 SVs may be lost, accounting for only 1.78% (1092/61322) of the total SVs, which has little effect on subsequent analysis.

4. In the annotation pipeline, it seems like no filtration of the predicted genes was performed. Is that so? For instance, do the annotations include gene models derived solely from ab-initio prediction? In my experience, tools like AUGUSTUS tend to over-annotate, so some filtration or QC of the annotations could be useful. One option would be using Annotation Edit Distance (AED) cutoff.

Response: We added some description of the filtering criteria used in the gene annotation process in the Methods section. For the gene models merged by Augustus, GeneWise, and PASA, we further filtered out incomplete gene models and gene models overlapping with repeats if the overlap ratio of CDS region more than 80%. For genes with CDS lengths less than 150 bp or less than 750 bp and 3 CDS, we used the Pfam database for validation. If no alignment result was obtained or the alignment coverage less than 25%, the gene model will be filtered out.

5. I am a bit skeptical of the phylogenetic analysis presented here. It is often difficult to infer a reliable phylogeny within a species due to incomplete lineage sorting and hybridization. That might be the reason for the relatively low bootstrap values. In addition, the inference was based on concatenation of a large number of genes, so a partitioned analysis might be required. It might be a good idea to:

- a) Reconstruct the phylogeny using SNP data and see if the resulting topologies are similar.
- b) Use a dimensionality reduction method like PCA to visualize the distances between genomes and see if the conclusions are the same.

Response: The bootstrap values displayed on the branches are percentages, and 100 means that the topology is 100% supported by 1000 bootstrap analyses. The bootstrap values for all branches are 74% or higher, indicating high support, and there is no case of low support (Fig. 1b). Principal component analysis (PCA) of 32 *Arabidopsis thaliana* ecotypes based on variable gene families also shows that the relict ecotypes are highly differentiated from other ecotypes, with Tibet-0 showing the greatest difference (Supplementary Figure 7), which is consistent with its position at the basal of the phylogenetic tree.

6. Regarding the analysis of the pan-TE: Can the authors relate this to the coding gene classification? E.g., do core genes tend to have less/more TEs etc.

Response: TEs tend to insert into variable genes (33.26%, 2561/7701), while the proportion of TE insertions in core genes is relatively smaller (15.28%, 3292/21545). And we added this sentence into the revised manuscript.

7. Regarding the SV analysis, it would be interesting to see if SVs can explain gene presence/absence patterns obtained in the annotation-based pan-genome.

Response: We found that 3,415 out of 7,701 (44.34 %) of variable gene families are affected by four types of SVs (> 50bp) and added this result into the revised manuscript.

8. As written above, it is preferential to have higher coverage for some accessions.

Response: Undeniably, increasing sequencing coverage may further improve the continuity of genome assembly, but based on the current contig length and number of contigs assembled, it can be seen that a 15X dataset is sufficient for high continuity assembly of the *A. thaliana* genome, and there is no significant decrease in continuity and completeness (BUSCO) observed (Fig. R1 and Supplementary Table 2, 6, 7).

9. The authors claim that some of the SV identified are the cause for some detected function or changes in expression (e.g., line 228-229), but this is more of a speculation, without any functional validation – can the authors functionally demonstrate that a given SV leads to differences in expression or function?

Response: We added in vivo dual-luciferase (Dual-LUC) activity assays of *KNAT3* and *CCRI* gene and demonstrated the effect of the insertions on gene expression (Fig. 2k, i and Fig. 3j, k). In addition, we further added functional validation experiments for the *WHI* and *HPCAI* genes, which confirmed the SVs contributes on gene expression as well as UVB and drought resistance (Fig. 4).

Minor

As written above, the writing can be improved. Here are some examples.

* Lines 6-7: better explain the limitations of a single reference genome.

Response: We modified this sentence into “However, previous studies based on single reference genomes and short-read sequencing data are restricted to detecting variable genes and large structural variation (SV) underlying local adaptation because of its short read length and high sequencing error rate.”

* Line 7: "chromosome-level" instead of "chromosomal"

Response: We have modified this.

* Line 8: I'm not sure all readers will be familiar with the term “relict ecotype” – can briefly explain. Also "including six relict..." would be better.

Response: The concept of relict ecotypes is well-known in *A. thaliana* ecotype-related research. In order to keep the Abstract part concise, we did not provide an explanation in this section, but added it to the Introduction part instead: “Some ecotypes of this species from Europe,

Africa, and Asia may have retained to be relict that are paraphyletic, while the majority of the Eurasian ones likely originated from one recent monophyletic expansion indicated by the previous genomic analyses. We therefore sampled six sparsely districted relict ecotypes...”.

* Line 9: "several thousand" - how many exactly?

Response: We have modified this sentence into “From these newly assembled genomes, we annotate 1,562 - 4,774 new genes or genes with structure variant in comparison to the previous reference genome.”

* Lines 11-12: "this species substantially expands its gene repertoire for local adaptation" - unclear, please rephrase.

Response: We modified this sentence into “The identified variable genes through pan-genomic analysis are mainly enriched in and associated with ecological adaptation, greatly expanding the gene pool associated with local adaptation in different ecotypes.”

* Line 15: "...was found to be specific...".

Response: We have modified this.

* Line 18: "...captured much of the missing heritability".

Response: We modified this into “supplement a proportion of missing heritability”.

* Line 25: "studies" instead of "research".

Response: We have modified this.

* Line 27: "complexity" - do you mean diversity?

Response: We have modified this.

* Line 40: is it really unknown if genetic variation contributes to local adaptation? I suspect this had been explored before.

Response: We modified this sentence into “Beyond SNPs and INDELS, there are only a few studies on whether both variable genes and large structural variations (SVs, often > 50 bp) contribute to ecological adaptation”.

* Line 54 - please expand a bit more on relict ecotypes and better explain why you decided to include these.

Response: We added some description of relict ecotypes and other ecotypes here and both of the two types are important: “Some ecotypes of this species from Europe, Africa, and Asia may have retained to be relict that are paraphyletic, while the majority of the Eurasian ones likely originated from one recent monophyletic expansion indicated by the previous genomic analyses. We therefore sampled six sparsely districted relict ecotypes...”.

* Line 70: "Chromosome-level genome assemblies and annotation of 38 ecotypes" (or break into two sections).

Response: We have modified this.

* Line 82: "contiguity" instead of "continuity".

Response: We have modified this.

* Line 85: "liftoff mapping" - not clear to everyone. Maybe use the general term "gene liftover" and explain the idea.

Response: We have modified this.

* Lines 87-90: the rationale and procedure are not clear. Please rewrite.

Response: We rewrite this sentence into “To maintain consistency with the Araport11 gene annotation, we prioritized retaining complete gene structures with gene lift-over mapping identity and coverage greater than 90%. We then added new annotated genes (based on ab initio prediction, homologous-protein-based prediction and transcriptome-based annotation) that did not overlap with existing gene annotations, including new genes or genes with structural variations (such as exon shifts) compared to Araport11.”

* Line 100: I suggest "... to construct a maximum likelihood (ML) phylogeny."

Response: We have modified this.

* Line 134. Change to: "...variable genes evolve more rapidly and may obtain new functions or more rapidly adapt to the environment.”

Response: Combining your suggestions and those of Reviewer 3, we have modified this sentence into “These results suggest that the function of core genes is relatively conservative while variable genes evolve more rapidly and may obtain new functions or more rapidly adapt to the environment, or the difference in Ka/Ks could simply be relaxed selection on non-core genes.”

* I didn't follow the difference between the content written in line 143 and then 147-148.

Response: Line 143 means the TE types in the pan-TE library and line 147-148 means the TE types in each genome. And we modified the sentence in 147-148 into “Among all TE categories, LTR-RTs and DNA transposons (such as terminal inverted repeats (TIR)) were the most abundant contents in different genomes”.

*Line 154 – It is unclear how exactly genes with TE insertion were defined

Response: We modified this sentence into “To evaluate the effect of TE insertion on gene expression, we compared the gene expression levels of genes with and without TE insertion (TE overlapped with gene region).”

* Line 181: "The specific sequences in each ecotype varied from 56.58 Kb to 8.45 Mb" - unclear. Do you mean novel sequences?

Response: Yes, we mean ecotype-specific novel sequences. We modified this sentence into “The novel sequences in each ecotype compare with the Col-0 reference varied from 56.58 Kb to 8.45 Mb...”

* I suspect that most potential readers are not familiar with the characteristics of the pan-graph. Few

sentences should be added as a preface to that section (line 176 onward) – what are the meanings of the nodes and the edges. More so, can we learn anything from the reported statistics regarding the graph’s properties? What can we learn from the fact that the graph has 457k nodes? Is this a lot? Does it provide any information? How is this compared to other species?

Response: As we mentioned in the Method part: “The fragments differing from the reference genome are displayed as different paths in the generated graphical fragment assembly (GFA) file. If two or more paths are connected between two fragments, they will form bubbles. Each bubble in the graph represents a structural variation.”. This is a routine statistic related to the number of SVs in graph pan-genome, and we added some descriptions in the main text. We modified this sentence into “The graph genome comprised a total of 243.27 Mb sequences with 468,168 nodes (the number of fragments of sequences) and 649,692 edges (the connections between nodes).”

*Line 186 Here and in line 108 – how exactly the plateau point was identified? By eye or using any statistical measure?

Response: The rule for platform points identification in line 108 is that after adding a new genome, the number of newly added gene families in the pan-genome is less than 1% of the total number of gene families. And we modified sentence at line 186 into “The pan-genome size increased with the number of genomes added.” as the Reviewer1 suggested.

*Line 194-5. Similarly, not much insights we can gain from the statistics given here – can these be compared to other species?

Response: Here is just a routine statistic on the proportion of genome length occupied by different types of SVs, which shows that divergent type of SVs affects the most regions, while deletion type of SVs affect the least regions.

*Line 238-246. What does it mean “missing heritability? What does it mean “SVs were found to contribute more shares of heritability”?

Response: Traditionally, the heritability of a phenotype is measured through familial studies using twins, siblings and other close relatives, making assumptions on the genetic similarities between them. Missing heritability describes the phenomenon that variance explained by fitting all the genetic markers is smaller than that estimated from familial studies. Here, we do not have a pedigree to obtain a robust estimate of heritability, missing heritability means kinship heritability is an underestimate, thereby, a fraction of the heritability is missing.

Our intention of “SVs were found to contribute more shares of heritability” was to describe the observation that SVs could explain more additive variance than SNPs when fitted together in a linear mixed model. One of the reasons underlying missing heritability is that the genetic markers were in imperfect LD with the causal variants. With the availability of graph-based pan-genome, we could genotype a large number of SVs that explain larger proportion of phenotypic variance for 48 analysed traits, suggesting a fraction of these SVs were either causal or could tag the causal variants better than SNPs, thereby, capturing more phenotypic variance.

We have revised the main text by saying “SVs were found to explain larger proportion of

phenotypic variance” to improve presentation. We hope our revision addressed your concerns and are open to your further suggestions.

* Line 519: the link doesn't work ("DOI not found"), so currently the data cannot be accessed.

Response: The genome assembly, genome annotation, pan-TE library, graph pan-genome, gene family and gene presence/absence matrices files have been deposited in the Figshare database (<https://dx.doi.org/10.6084/m9.figshare.21673895>). The private link for reviewers is <https://figshare.com/s/60f2b253e4648767ddf2>.

* Line 688-9: "Branches with 100 bootstrap values are not shown" - do you mean less than 100? but the branches in figure 1b show lower numbers. It is written in the methods that 1000 BS were used, so I'm confused. Also "green branches" - it looks blue to me.

Response: We means branches without displayed bootstrap values were 100% supported (with 1000 ultrafast bootstrap analyses). Now we show the bootstrap values (%) of all the branches in Fig. 1b. And we have modified “green” into “blue”.

Reviewer #3 (Remarks to the Author):

The authors sequenced and assembled the genomes of 38 *A. thaliana* accessions. This is an impressive dataset and will likely be of considerable value to the research community, though the paper provided few novel insights.

Mostly, I was left unconvinced that data support claims regarding the adaptive value of the variation described. Certainly, SV, TE, etc contribute to adaptive differentiation across climate gradients but the results presented in the paper to explore this were not very compelling.

For CCR1 and KNAT3, the authors showed that the accession with an SV had lower expression. No experiments were conducted that confirm the causality of that SV on expression, nor on the adaptive value in relation to the environment. As such phrasing like “Because of this insertion, the gene expression level in Tibet-0 was significantly reduced compared with other ecotypes without this insertion” (line 229) are not warranted.

Response: We added in vivo dual-luciferase (Dual-LUC) activity assays of *KNAT3* and *CCR1* gene and demonstrated the effect of the insertions on gene expression (Fig. 2k, i and Fig. 3j, k). In addition, we further added functional validation experiments for the *WHI* and *HPCA1* genes, which confirmed the SVs contributes on gene expression as well as UVB and drought resistance (Fig. 4).

Fig.5d. Why is the y-axis so high?

Response: We have modified this.

Line 116-117: what is a “basic” biological process? All of the processes listed are quite complex. Can the authors find a more precise word?

Response: We modified this sentence into “Gene ontology (GO) term enrichment analysis revealed that the core genes were enriched in basic, critical functions such as flower

development, RNA binding, transcription regulate, transport and cellular homeostasis, which suggests that the core genes are mainly involved in maintaining the basic activities of *A. thaliana*.” which similar to the description in sorghum pan-genome (Tao. Y. et. al., 2021).

133:134: Authors should also mention that the difference in Ka/Ks could simply be relaxed selection on non-core genes (rather than diversification due to selection for novel functions or adaptation).

Response: Combining your suggestions and those of the other Reviewer, we have modified this sentence into “These results suggest that the function of core genes is relatively conservative while variable genes evolve more rapidly and may obtain new functions or more rapidly adapt to the environment, or the difference in Ka/Ks could simply be relaxed selection on non-core genes.”

The authors observed bias in the distribution of SV with relation to gene features. This is an interesting result of the paper. However, it is unclear whether the discussion of this is implying this distribution reflect bias in the insertion site or bias in selection on “random” TE insertions.

Response: The initial TE insertions may be random and their retentions are selected due to the regulation of gene expression with adaptive roles

169: Is this a mechanistic explanation or an adaptive explanation?

Response: This is just a speculation based on the number of TEs inserted in different regions of the genome.

173:-174. Again, can't tell if this is a hypothesis about preference in insertion sites or an adaptive hypothesis.

This would be a good paper to check out <https://www.ncbi.nlm.nih.gov/pmc/articles/PMC6668482/> Shows clearly the basis of biased insertion by TE which likely is relevant for these findings.

Response: The initial TE insertions may be random and their retentions are selected due to the regulation of gene expression with adaptive roles.

It is very unclear what heritability is being measured. The authors talk about partitioning phenotyping variance, but it was not clear what phenotype was even measured. The methods section does not mention any traits, it only (briefly) describes the analyses.

Response: We have clarified this by revising the methods sections to include more details on the analyzed phenotype and the analysis methods. In this revision, 61 traits, including 19 climate variables, global UV-B radiation level and elevation data as well as two flowering time traits measured at 10°C and 16°C, and 38 ionomics phenotypes were used to evaluate the role of SVs in dissecting the genetic basis of adaptive traits. We hope our revision improved the presentation and are open to your further suggestions.

In any case, it seems hardly accurate that the SV explained “the missing heritability” as stated in the abstract or “supplement a large proportion of heritability” (line 299). The improvement with SV was marginal: 0.03% greater than SNPS alone.

Response: We fully agree with you that the improvement in capturing missing heritability was marginal (1.18 % in the revised manuscript). Despite this, we do find a large number of

association peaks in SV- GWAS that were missed in SNP-GWAS. Therefore, we think leveraging SV in dissecting the genetic basis of adaptive traits is a valuable approach.

Given these results, we have tune it down in our revised manuscript by saying “SV supplement a proportion of heritability and uniquely associated with a number of adaptive traits”. We hope our revision addressed your concerns and are open to your further suggestions.

The authors should mention and cite <https://www.nature.com/articles/s41467-020-14779-y>

Response: We cited this reference article in the revised manuscript.

The link to data available <https://dx.doi.org/10.6084/m9.figshare.21673895> doesn't work. It would be helpful so we can review the quality and accessibility of the data.

Response: The genome assembly, genome annotation, pan-TE library, graph pan-genome, gene family and gene presence/absence matrices files have been deposited in the Figshare database (<https://dx.doi.org/10.6084/m9.figshare.21673895>). The private link for reviewers is <https://figshare.com/s/60f2b253e4648767ddf2> .

The paper is written well and I think this could be a very highly cited and valuable paper given the magnitude of the data, but the follow-up analyses presented to glean some ecological or biological insights from the data do feel rather underwhelming.

Reference

Tao, Y., Luo, H., Xu, J., Cruickshank, A., Zhao, X., Teng, F., ... & Mace, E. (2021). Extensive variation within the pan-genome of cultivated and wild sorghum. *Nature Plants*, 7(6), 766-773.

Reviewers' Comments:

Reviewer #1:

Remarks to the Author:

This revised manuscript has added necessary details and much analyses especially the SV-based GWAS. The authors also have addressed my concerns. Overall, I am satisfied with this revision. Here I only have some minor comments for consideration.

Minor comments

1. Line 33 It might be better to use "Telomere-to-telomere" instead of "T2T"
2. Line 85-86, citation for BUSCO
3. Line 104, citation or webpage link for eggNOG
4. Line 131, citation for BIOCLIM environmental variables
5. For Fig. 3g and Fig. 4a, h, it would be helpful for understanding to add some information like the coordinates or distance/length of the gene/exons and TE insertion. By the way, the main text says these insertions are in the promoter regions. Do you mean the 1kb upstream of gene Transcription Start Site or other definition? Please clarify this.

Reviewer #3:

Remarks to the Author:

I remain enthusiastic about the value of the data generated in this study and I appreciate the authors' effort to address previous comments. They have made significant improvements, especially with respect to making the data more available for community use. This will increase the impact of their work. I also appreciate the addition of experiments to test hypotheses about specific SVs functionally. This definitely aids in showing how pan-genomes can yield new discoveries.

Regarding the distribution of TE in relation to gene features, I think the authors need to further clarify whether they are suggesting bias in the insertion site or bias in selection on "random" TE insertions. I think this is a critical point that should be addressed clearly.

While discussing the intriguing distribution of TE insertions, I appreciate that the authors have considered the role of selection in the retention of these insertions. However, another crucial factor that could potentially explain this distribution is the tendency of TEs to insert themselves in certain chromatin contexts. This is a well-documented phenomenon in the literature, and considering it in their interpretation could provide a more comprehensive understanding of their findings.

I highly recommend an in depth read of Quadrana et al 2019, which demonstrated, among other things, "an essential role of the histone variant H2A.Z in the preferential integration of Ty1/copia retrotransposons within environmentally responsive genes and away from essential genes."

This (and other observations, including preferential insertion into promoters for TE families) seems quite relevant to this and other statements in this manuscript regarding the TE distribution:

Lines 169-183

"TEs tend to insert into variable genes (33.26%, 2561/7701), while the proportion of TE insertions in core genes is relatively smaller (15.28%, 3292/21545)...TEs were more likely to insert into the upstream regions of the genes with functional enrichments with habitat adaptation (Supplementary Figure 13). The initial TE insertions may be random and their retentions are selected due to the regulation of gene expression with adaptive roles. "

Preferential TE insertion bias does not necessarily contradict their current hypothesis; in fact, it could be that both the TE's insertion preference and selection contribute to the observed distribution. I

recommend the authors incorporate this viewpoint and explicitly present these two hypotheses - TE insertion tendencies and selection pressures - as non-mutually exclusive factors influencing the distribution of TE insertions. This addition would enrich the discussion and interpretation of their results, without detracting from the overall message of their work.

REVIEWER COMMENTS

Reviewer #1 (Remarks to the Author):

This revised manuscript has added necessary details and much analyses especially the SV-based GWAS. The authors also have addressed my concerns. Overall, I am satisfied with this revision. Here I only have some minor comments for consideration.

Minor comments

1. Line 33 It might be better to use “Telomere-to-telomere” instead of “T2T”

Response: We have modified T2T into telomere-to-telomere (T2T).

2. Line 85-86, citation for BUSCO

Response: We have modified this.

3. Line 104, citation or webpage link for egglog

Response: We have modified this.

4. Line 131, citation for BIOCLIM environmental variables

Response: We have modified this.

5. For Fig. 3g and Fig. 4a, h, it would be helpful for understanding to add some information like the coordinates or distance/length of the gene/exons and TE insertion. By the way, the main text says these insertions are in the promoter regions. Do you mean the 1kb upstream of gene Transcription Start Site or other definition? Please clarify this.

Response: We added the distance of the TE to the transcription start site (TSS) in Fig. 3g and Fig. 4a, h. And we added the definition of the promoter region: “upstream 2k from the transcription start site (TSS)” in Line 229.

Reviewer #3 (Remarks to the Author):

I remain enthusiastic about the value of the data generated in this study and I appreciate the authors' effort to address previous comments. They have made significant improvements, especially with respect to making the data more available for community use. This will increase the impact of their work. I also appreciate the addition of experiments to test hypotheses about specific SVs functionally. This definitely aids in showing how pan-genomes can yield new discoveries.

Regarding the distribution of TE in relation to gene features, I think the authors need to further clarify whether they are suggesting bias in the insertion site or bias in selection on “random” TE insertions. I think this is a critical point that should be addressed clearly.

While discussing the intriguing distribution of TE insertions, I appreciate that the authors have considered the role of selection in the retention of these insertions. However, another crucial factor that could potentially explain this distribution is the tendency of TEs to insert themselves in certain chromatin contexts. This is a well-documented phenomenon in the literature, and considering it in their interpretation could provide a more comprehensive understanding of their findings.

I highly recommend an in depth read of Quadrana et al 2019, which demonstrated, among other things, “an essential role of the histone variant H2A.Z in the preferential integration of Ty1/copia retrotransposons within environmentally responsive genes and away from essential genes.”

This (and other observations, including preferential insertion into promoters for TE families) seems quite relevant to this and other statements in this manuscript regarding the TE distribution:

Lines 169-183

“TEs tend to insert into variable genes (33.26%, 2561/7701), while the proportion of TE insertions in core genes is relatively smaller (15.28%, 3292/21545)...TEs were more likely to insert into the upstream regions of the genes with functional enrichments with habitat adaptation (Supplementary Figure 13). The initial TE insertions may be random and their retentions are selected due to the regulation of gene expression with adaptive roles. “

Preferential TE insertion bias does not necessarily contradict their current hypothesis; in fact, it could be that both the TE's insertion preference and selection contribute to the observed distribution. I recommend the authors incorporate this viewpoint and explicitly present these two hypotheses - TE insertion tendencies and selection pressures - as non-mutually exclusive factors influencing the distribution of TE insertions. This addition would enrich the discussion and interpretation of their results, without detracting from the overall message of their work.

Response: We are highly grateful for your constructive and thoughtful comments. We have modified the sentences in Line 183-189 into “The observed TE insertion distribution bias here may be due to following two reasons: 1) The initial TE insertions may be random, and their retentions are selected due to the regulation of gene expression with adaptive roles. 2) the TEs targets have specific chromatin signatures. For example, previous published study demonstrated that the histone variant H2A.Z played an essential role in the preferential integration of Ty1/Copia retrotransposons within environmentally responsive genes and away from essential genes. These two hypotheses may jointly and non-exclusively affect the distribution of TE insertions in *A. thaliana*.” and cited the reference of Quadrana et al 2019.

Reviewers' Comments:

Reviewer #3:

Remarks to the Author:

The authors have addressed all of my concerns. Nice job!